# Learning Self-Critiquing Mechanisms for Region-Guided Chest X-Ray Report Generation

**Sixing Yan**[1,2]*, **Ziao Wang**[2], **Kejing Yin**[2]†, **William Cheung**[2], **Ka Chun Cheung**[3] **& Simon See**[3]
[1] Postdoctoral Research Station, Guangxi Rural Commercial United Bank Co., Ltd
[2] Department of Computer Science, Hong Kong Baptist University
[3] NVIDIA AI Technology Center, NVIDIA Corporation

## Abstract

Clinically accurate and interpretable automatic radiology reporting requires reliably grounding the identified abnormalities with the corresponding regions located in the radiology image. In this paper, we propose to introduce self-critiquing mechanisms into the automatic report generation process to ensure the identified abnormalities can reliably grounded before they are reported. Instead of adopting LLM-based reasoning to implement the self-critiquing mechanisms which will incur high inference cost in test time, we propose a novel Radiology Self-Critiquing Reporting (RadSCR) model framework which allows multi-faceted mechanisms to be learned end-to-end to identify and verify some hypothesized abnormality regions by comparing with i) alternative abnormalities, ii) alternative patients' X-ray images, and iii) potential false negatives. The self-critiqued abnormality proposals are then integrated using a retrieval-based approach to generate the final report. Our experimental results show that RadSCR can outperform the state-of-the-art report generation methods in terms of clinical accuracy by a large margin, with improved reliability of abnormality localization.

## 1 Introduction

Automated radiology image reporting aims to reduce radiologists' workloads on report preparation. Recent development of deep learning models for generating X-ray reports has shown continuous improvement on clinical accuracy Chen et al. (2020); Yan et al. (2023); Wang et al. (2024c). Yet, how to reliably grounding a generated report with the regions of the abnormalities identified in the images remains open, which is important as this is what radiologists carry out in practice. In this paper, we propose to incorporate self-critiquing mechanisms into deep learning models for generating X-ray reports so that the reliable grounding of the abnormality findings can be established.

Grounding radiology images with abnormality findings using deep learning models is non-trivial as large-scale annotations of abnormality regions are still lacking. Some recent works explored anatomy-awareness by making reference to detected anatomical parts (e.g., lung, heart, etc.) in the image for grounding the findings, resulting in higher accuracy and better interpretability (Tanida et al., 2023; Li et al., 2024; Dalla Serra et al., 2023; Yan et al., 2024). In practice, more fine-grained abnormality regions are generally preferred for grounding. Also, carefully examining potential abnormality regions is often unavoidable if a reliable radiology report is to be prepared.

Self-critiquing is commonly adopted by medical professionals to reduce the chance of making diagnosis mistakes. In the context of report preparation, it refers to the process where radiologists identify and validate the potential abnormality regions on the X-ray images before findings are concluded. Existing automatic radiology reporting models are mostly trained based on statistical correlations between regions and paired sentences (Fallahpour et al., 2025; Gai et al., 2024; Fan et al.,

---

*This work was done when Sixing Yan was a Postdoctoral Fellow in the Department of Computer Science, Hong Kong Baptist University.
†Correspondence author: Kejing Yin (cskjyin@comp.hkbu.edu.hk).

2025), resulting in unavoidable hallucinations. The idea of self-critiquing is still under-explored, except for a few works on visual question answering (Cheng et al., 2025; Wu & Mooney, 2019).

We argue that reliable radiology report generation requires "multi-faceted" self-critiquing mechanisms for establishing reliable grounding of potential abnormalities. While large language model-based paradigms like chain-of-thought have recently been explored to introduce test-time reasoning to alleviate hallucination (Wu & Mooney, 2019; Cheng et al., 2024; Cocchi et al., 2025), we consider alternatives as LLM-based reasoning typically generates a long chain of "thinking" tokens during inference and incurs high test-time cost (Huang et al., 2025; Geiping et al., 2025). Also, deploying LLMs for applications with a low-resource environment is non-trivial. Our idea is to incorporate multi-faceted self-critiquing mechanisms into the model architecture to be learned during the training, without requiring test-time scaling.

To this end, we propose a novel **Rad**iology **S**elf-**C**ritiquing **R**eporting (RadSCR) model framework which adopts a region-guided chest X-ray report generation paradigm with self-critiquing mechanisms incorporated to mimic the self-critiquing thinking process of radiologists for enhancing the report's reliability. RadSCR first identifies an initial set of fine-grained *visual proposals*, each represented by a triplet of abnormality region, abnormality label and the corresponding visual features. Self-critiquing is then realized by cross-checking the hypothesized visual proposals to see if their visual features are *distinct* and *relevant* enough for the associated abnormalities. In particular, it explores *alternative abnormalities* and *alternative patients' X-ray images*, and then further takes a holistic view of the image to double-check the *possibility of missing abnormalities*. The visual proposals "discounted" by the possible alternatives are considered together for retrieving appropriate sentences of abnormality findings from a report repository to be integrated by an LLM decoder to generate the final report. We carried out comprehensive experiments to evaluate the effectiveness of RadSCR using a variety of datasets including MIMIC CXR, ReXGradient, and IU X-ray. Our experimental results demonstrate that RadSCR outperforms all the state-of-the-art report generation baselines by a large margin, with improved localization of abnormality regions for grounding the findings. The main contributions of the proposed RadSCR include:

- providing an automatic radiology reporting methodology guided by abnormality regions for more fine-grained grounding of abnormality findings;

- introducing self-critiquing mechanisms into a deep model architecture for more reliable grounding without the need to introduce LLM-based reasoning in test time;

- demonstrating via comprehensive empirical evaluation the effectiveness of introducing self-critiquing mechanisms to achieve clinically accurate radiology X-ray reporting.

## 2 RELATED WORKS

**Grounded Radiology Report Generation** The grounding of the generated findings of the report within the relevant regions on radiology images is important for medical image understanding and diagnosis (Bannur et al., 2024). Various well-designed attention mechanism modules have been proposed to locate abnormality region of interests (ROIs) for X-ray report generation. Wang et al. (2024a) proposed to use class activation mapping (CAM) (Zhou et al., 2016) to guide the visual attention module to identify regions of abnormalities, where vision-weighted maps are obtained from a multi-abnormality classifier head topped at the visual encoder. Alternatively, the anatomy-awareness approach tries to locate the anatomical parts relevant to the findings generated for grounding. RGRG (Tanida et al., 2023) and ORGAN (Hou et al., 2023b) use a shared visual extractor to detect the regions of the anatomical parts and then generate the report accordingly. BoxMed-RL (Jing et al., 2025), MedPromptX (Shaaban et al., 2024) and MAIRA-2 (Bannur et al., 2024) learn to annotate the anatomical regions and detect the possible pathology labels, followed by report generation. However, localizing anatomical parts is not precise enough for the grounding purpose.

**Radiology Reasoning in Visual Question Answering** Radiology reporting typically involves a multi-step diagnostic process to identify and locate abnormalities revealed in the image. Recent advances in LLM-based reasoning approaches use chain-of-thought (CoT) to represent the process for report generation. For instance, MedCoT (Liu et al., 2024) incorporates several LLMs as hierarchical experts by CoT, where each expert's output is further verified by a subsequent expert. MedRAX (Fallahpour et al., 2025) decompose radiology image diagnosis into a sequence of tasks

and use multiple pre-trained models as agents to solve each task. ChestX-Reasoner (Fan et al., 2025) further decomposes each diagnostic finding of the report into a step-by-step CoT where each CoT contains a textual description, anatomical region, and expert-labeled clinical notes. Instead of using CoT, RECAP (Hou et al., 2023a) and ORGAN (Hou et al., 2023b) implement diagnostic reasoning by finding a proper graph walk in a pre-constructed knowledge graph of clinical findings. Our proposed RadSCR does not employ reasoning token generation in test time as CoT and achieves self-critiquing by incorporating that directly into the model architecture. In the literature, there exist some recent works which also take the "what-if" approach as RadSCR for more reliable radiology report generation. PGFC (Mahmood et al., 2025) uses a fact-checking model to determine whether a pair of a clinical finding and an anatomical region match with each other or not. CoFE (Li et al., 2025) creates counterfactual explanations by replacing patches on an X-ray image until the diagnosis changes for contrastive learning, where localization and grounding are not considered. In contrast, our proposed RadSCR critiques the predicted abnormality location by considering alternative abnormality and alternative image, and false negative checking; and use these critiquing to enhance the sentence retrieval reliability for report generation without generating counterfactual explanations.

**Weakly Supervised Abnormality Localization** Lacking large-scale annotation of abnormality regions in X-ray images makes supervised learning of abnormality localization difficult. Weakly-supervised learning approaches for abnormality localization have been investigated. Attention mechanisms are learned to attend regions of abnormalities, where abnormalities are then classified based on the visual features of these regions (Li et al., 2018). Anatomical areas can also be used to restrict potential regions for subsequent localization (Yu et al., 2022). With a similar idea, some coarse-grained abnormal regions can first be grounded before localizing the regions of specific abnormalities (Ouyang et al., 2020; Wang et al., 2024b). In general, how to precisely localize the regions (not to over-cover or over-look) for grounding the report generation remains open.

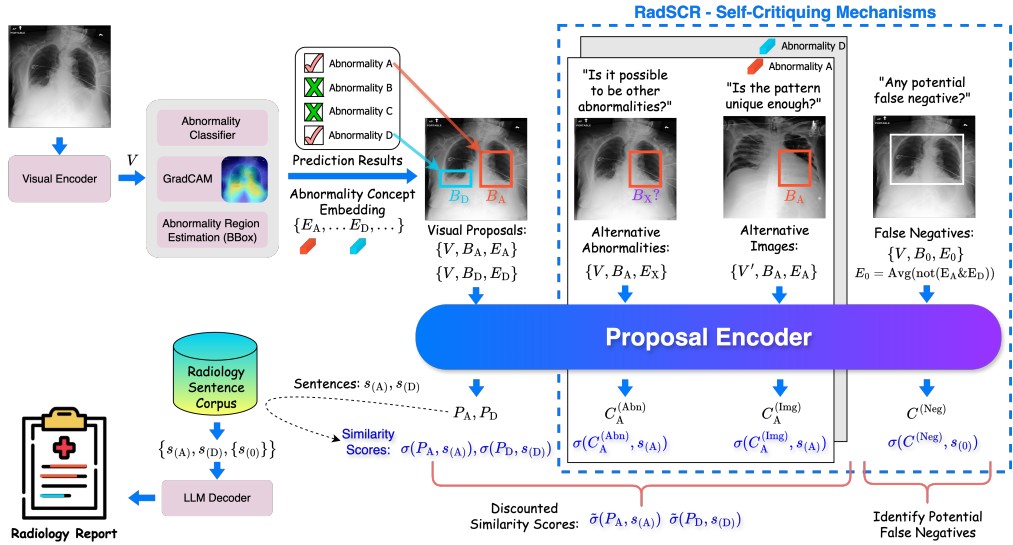

Figure 1: An overall model architecture of the proposed RadSCR for radiology report generation.

## 3 METHODS

Given a radiology image $I$, our proposed **Rad**iology **S**elf-**C**ritical **R**eporting (RadSCR) framework generates a radiology report $R$ with findings of abnormalities grounded with abnormality regions identified in $I$. First, a set of visual proposals of abnormalities is hypothesized. To obtain the visual proposals, an X-ray image $I$ encoded as the visual features $V$ is first fed to a classifier to predict the presence of a set of $\mathcal{N}$ abnormalities. The potential region of the $m$-th predicted positive abnormality, represented as a bounding box $B_m$, can be located using Grad-CAM (Selvaraju et al., 2017). For each abnormality, a concept embedding $E_m$ is to be learned. A region-based visual

proposal of the $m$-th abnormality is thus denoted as a triple $(V, B_m, E_m)$, and represented as:

$$P_m = \text{PropEncoder}(V, B_m, E_m). \tag{1}$$

Then, the hypothesized visual proposals are critiqued for their *distinctiveness* and *relevancy* using multi-faceted self-critiquing mechanisms. Three mechanisms are introduced to critique the visual proposals $\{P_m\}$ by proposing i) alternative abnormalities, ii) alternative patient X-ray images, and iii) potentially missing abnormalities, represented respectively as: $\{C_m^{(\text{Abn})}\}$, $\{C_m^{(\text{Img})}\}$, and $C^{(\text{Neg})}$. Each hypothesized visual proposal $P_m$ is considered together their corresponding alternative visual proposals $C_m^{(\text{Abn})}$ and $C_m^{(\text{Img})}$ to retrieve relevant sentences from the report repository, denoted as $Q_m$. $C^{(\text{Neg})}$ is used to retrieve additional sentences of potential false negatives which could be missed in the localization step, denoted as $U$. They are then fed to an LLM decoder with a pre-defined prompt for sentence aggregation and final report generation. The process is denoted as:

$$R = \text{Generate}(\{Q_m\}, U); \quad \{\{Q_m\}, U\} = \text{Retrieve}(\{(P_m, C_m^{(\text{Abn})}, C_m^{(\text{Img})})\}, C^{(\text{Neg})}). \tag{2}$$

Fig. 1 provides an overview of the RadSCR framework, with details to be presented.

## 3.1 CONSTRUCTING RADIOLOGY VISUAL PROPOSALS

We represent the visual features $V \in \mathbb{R}^{\mathcal{HW} \times \mathcal{D}}$ of an X-ray image $I$ as a patch map $\mathcal{H} \times \mathcal{W}$ with $\mathcal{D}$-dimensional features encoded by a visual encoder, denoted as $V = \text{VisEncoder}(I)$.

**Initial Abnormality Prediction** The presence of the $m$-th abnormality is predicted by feeding the average visual features of $V$ over the dimension of $\mathcal{HW}$ to a fully connected network $\text{FCN} : \mathbb{R}^{\mathcal{D}} \to \mathbb{R}^{\mathcal{N}}$, where the predicted probability is computed as:

$$p_m = \text{Sigmoid}(\text{FCN}(\text{AvgPool}(V))). \tag{3}$$

The $\mathcal{M}$ abnormalities with $p_m > 0.5$ form the set of potential (positive) abnormalities in $I$.

**Abnormality Region Localization:** The relevant regions on the image $I$ associated with each predicted positive abnormality are then to be localized. As conventional object detectors do not generalize well to out-of-distribution (OOD) abnormalities, we leverage the class activation mapping (CAM) to compute the pixel-level saliency on the image $I$ for each predicted positive abnormality Fu et al. (2020), with the saliency map denoted as $W_m = \text{CAM}(\text{VisEncoder}, I, m)$ (as shown in Fig. 1). Then, a set of corresponding bounding boxes, denoted as $BBoxes = \{b_{m,1}, b_{m,2}...\}$, is estimated based on $W_m$ using the objectness estimation algorithm proposed by Cheng et al. (2014), where $b_{m,*}$ is a patch-level mask sharing the same size of $I$ with the patches set as 1 to indicate the presence of the $m$-th abnormality, or 0 otherwise. The *localized region* of the $m^{th}$ abnormality $B_m \in \mathbb{R}^{\mathcal{HW}}$ is then formed by the union of the bounding boxes in $BBoxes$, given as:

$$B_m = \text{Map}\big(\text{MaxPool}(\{b_{m,1}, b_{m,2}, ...\})\big). \tag{4}$$

**Region-based Visual Proposal Encoding:** We conceptualize the $\mathcal{N}$ abnormalities and the common chest X-ray background using $(\mathcal{N}+1)$ global concept representations $E \in \mathbb{R}^{(\mathcal{N}+1) \times \mathcal{D}}$ (to be learned as detailed in Section 3.3). $\{E_m \in \mathbb{R}^{1 \times \mathcal{D}}\}_{m=1}^{\mathcal{N}}$ corresponds to the concept representations of the $\mathcal{N}$ abnormalities, and $E_{\mathcal{N}+1} \in \mathbb{R}^{1 \times \mathcal{D}}$ is the padding representation of the background.

The region-based visual proposal for the $m$-th positive abnormality $(V, B_m, E_m)$ is represented as: $P_m = \text{PropEncoder}(V, B_m, E_m)$. We implement the proposal encoder by first obtaining the spatial-aware abnormality representation $F_m$ based on abnormality region mask $B_m$ with the concept representation $E_m$, and then concatenating it with the visual features $V$ at the patch level to obtain an abnormality proposal representation $P_m \in \mathbb{R}^{\mathcal{HW} \times \mathcal{D}}$, given as :

$$P_m = \text{FFN}(V \oplus F_m); \quad F_m = \text{Embedding}(B_m, E_m), \tag{5}$$

where $\text{Embedding}()$ replaces the slots of abnormality localization in $B_m$ with $E_m$ and the remaining slots with padding embedding $E_{\mathcal{N}+1}$, and $\text{FFN}(\cdot)$ is a two-layer feed-forward network.

## 3.2 LEARNING SELF-CRITIQUING MECHANISMS

To ensure the reliability of the visual proposals, we mimic the thinking process of radiologists to re-examine the proposals' correctness by considering i) alternative abnormalities, ii) alternative patient

X-ray image, and iii) potential false negative abnormalities. With reference to each visual proposal $(V, B_m, E_m)$, we modify different components to form additional proposals for the critiquing.

**Critiquing Based on Alternative Abnormalities** For chest X-ray images, some abnormalities with similar appearance are hard to differentiate. To self-critique if a positive abnormality is distinct enough as compared to other abnormalities, we compute the visual proposal of an alternative abnormality $(V, B_m, E'_m)$, where $E'_m$ can be the concept representation of an abnormality selected from the predicted negative abnormalities or those predicted positive but in some areas not covered by $B_m$. The corresponding representation is given as:

$$C_m^{(\text{Abn})} = \text{PropEncoder}(V, B_m, E'_m). \tag{6}$$

**Critiquing Based on Alternative X-ray Images** To check if a visual proposal is specific enough and relevant for diagnosing the abnormality among patients with different abnormalities, we form another type of alternative proposal for critiquing by replacing the visual features of the hypothesized visual proposal $(V)$ with those of a randomly selected image from other patients $(V')$. The representation of the critique is obtained by:

$$C_m^{(\text{Img})} = \text{PropEncoder}(V', B_m, E_m). \tag{7}$$

**Critiquing by Considering Potential False Negatives:** As all visual proposals are hypothesized using fine-grained localization, abnormalities characterized by more holistic features could be "overlooked". To double-check for such false negatives, we leverage the global concept representations of the predicted negative abnormalities to make an additional complementary visual proposal. Localizing the predicted negatives is non-trivial as reliable bounded boxes are absent. Instead, we compute an overall complementary visual proposal by taking an average pooling of the concept representation of all the predicted negatives, given as $E_0 = \text{AvgPool}(\{E_m | p_m < 0.5, m \in [1, \mathcal{N}]\})$. We then associate it with a region by aggregating the bounding boxes of the major anatomical parts in the Chest region based on the automatic detection tool (Seibold et al., 2023) if not annotated, denoted as $B_0$. The visual proposal is thus denoted as $(V, B_0, E_0)$, and represented as:

$$C^{(\text{Neg})} = \text{PropEncoder}(V, B_0, E_0). \tag{8}$$

## 3.3 Retrieval-based Report Generation With Self-Critiquing

With an X-ray image represented as $(\{(P_m, C_m^{(\text{Abn})}, C_m^{(\text{Img})})\}, C^{(\text{Neg})})$ we adopt the retrieval-based approach for report generation (Endo et al., 2021; Yang et al., 2021; Ranjit et al., 2023; Yan et al., 2024). Sentences of relevant findings matched with the image are first retrieved from a repository of radiology reports and then combined using an LLM to generate the final report.

**Representations of Sentence and Prototype:** We represent a sentence annotated with the $m$-th abnormality in a report as $s_{(m)}$. For each report, we concatenate all sentences of the same abnormality into one to ease the subsequent retrieval. To allow robust retrieval of $s_{(m)}$ based on the visual proposal $P_m$, we argue that it is important for not only $s_{(m)}$ aligning well with $P_m$, but also its higher-level clinical concepts (called "prototypes" in the following sections). For each abnormality, we assume $\mathcal{K}$ prototypes, represented as $O := \{o_k\}_{k=1}^{\mathcal{K}}$. We learn the sentence representation $s_{(m)}$ and the prototype representation $o_{pt(s_{(m)})}$ so that they are close to $P_m$ over all the sentences in the repository, where $pt(s)$ gives the index of $s$'s associated prototype.

To compute $s_{(m)}$, we first apply pre-trained ClinicalBERT (Yan & Pei, 2022) to obtain $s_{bert} \in \mathbb{R}^{l \times \mathcal{D}}$ where $l$ is the sentence length, and then cross-attention $\text{Attn}_\text{x}()$ between $s_{bert}$ with the concept representation $E_m$ to pick up the associated semantics. Then, self-attention $\text{Attn}_\text{s}()$ with average pooling is used to obtain the sentence representation $s_{(m)} \in \mathbb{R}^{1 \times \mathcal{D}}$:

$$s_{(m)} = \text{AvgPool}(\text{Attn}_\text{s}(T'_m, T'_m, T'_m)) \quad \text{where} \quad T'_m = \text{Attn}_\text{x}(s_{bert}, E_m, E_m). \tag{9}$$

To derive the prototypes $O$, we first apply $\mathcal{K}$-means to the TF-IDF representation of the sentences in the report repository. The sentence-cluster association is then fixed. The representation of the $k$-th prototype is initialized by applying average pooling to the representations of the associated sentences, denoted as $o_k = \text{AvgPool}(\{s : pt(s) = k\})$, and will be optimized during training.

**Retrieving Relevant Sentences with Self-Critiquing and Report Generation:** Given the visual proposal $P_m$, relevant sentences are retrieved among those annotated with the $m^{th}$ abnormality to support the report generation. To support more robust retrieval of relevant sentences, we compute the similarity score between the visual proposal $P_m$ and a sentence $s_{(m)}$ by considering also the sentence's prototype, given as:

$$\sigma(P_m, s_{(m)}) = P_m \odot s_{(m)} + \alpha_1 P_m \odot o_{pt(s_{(m)})}, \tag{10}$$

where $\odot$ is the dot product and $\alpha_1$ is the importance weight of the prototype-based similarity. For enhancing reliability, we incorporate the alternative visual proposals $C_m^{(\text{Abn})}$ and $C_m^{(\text{Img})}$ to suppress $P_m$ and discount the similarity score as:

$$\tilde{\sigma}(P_m, s_{(m)}) = \sigma(P_m, s_{(m)}) - \alpha_2\big(\sigma(C_m^{(\text{Abn})}, s_{(m)}) + \sigma(C_m^{(\text{Img})}, s_{(m)})\big), \tag{11}$$

where $\alpha_2$ is the importance weight of the alternative proposals. The top-$\mathcal{M}$ sentences retrieved based on the discounted similarity score $\tilde{\sigma}(P_m, s_{(m)})$ form the candidate set of sentences $\{Q_m\}_{m=1}^{\mathcal{M}}$.

For the complementary visual proposal $C^{(\text{Neg})}$, the use of prototypes is not needed as the predicted negatives are aggregated together in our formulation. The similarity score between $C^{(\text{Neg})}$ and the sentences corresponding to $E_0$ (denoted as $s_{(0)}$) can be computed by $\sigma(C^{(\text{Neg})}, s_{(0)}) = C^{(\text{Neg})} \odot s_{(0)}$. The top-$(\mathcal{N}-\mathcal{M})$ sentences based on $\sigma(C^{(\text{Neg})}, s_{(0)})$ form the complementary set of candidate sentences denoted as $\{U_n\}_{n=1}^{\mathcal{N}-\mathcal{M}}$.

To generate the final report $R$, an LLM is adopted to integrate the retrieved results using a prompt:

$$R = \text{LLM}\big(Prompt(\{Q_m\}_{m=1}^{\mathcal{M}}, \{U_n\}_{n=1}^{\mathcal{N}-\mathcal{M}})\big). \tag{12}$$

In our experiment, the *Prompt* is designed so that all sentences in $Q$ are expected to be used for report generation, while those in $U$ are only used if they do not contradict $Q$ (as shown in Fig. 3).

### 3.4 Loss Function for Model Learning

The RadSCR model $\mathbf{M}$ is designed with the following learnable components: the visual encoder VisEncoder(), the initial abnormality predictor FCN(), the abnormality concept representations $\{E_m\}$, the visual proposal encoder PropEncoder(), the sentence attention mechanisms $\text{Attn}_s()$ and $\text{Attn}_x()$, and the prototype representations $O$. For model learning, we define an objective function with a set of loss terms to achieve reliable retrieval.

**Loss for Visual-Language Alignment:** For each image-report pair (indexed by $i$) in the training batch $\mathcal{B}$, each underlying visual proposal $P_m^i$ should be close to the positive samples which are the ground-truth sentence $s_{(m)}^i$ and the prototype $o_{pt(s_{(m)})}^i$, but far from the negative samples containing sentences of: a) other abnormalities $\{s_{(\neg m)}^i\} := \{s_{(j)}^i\}_{j \in \{1...\mathcal{M}\}\setminus m}$; b) same abnormality but in different reports $\{s_{(m)}^{\neg i}\} := \{s_{(m)}^j\}_{j:=\{1...|\mathcal{B}|\}\setminus i}$; and c) same abnormality but with different prototypes $\{o_{\neg pt(s_{(m)}^i)}\} := \{o_k\}_{k \in \{1..\mathcal{K}\}\setminus pt(s_{(m)}^i)}$. The loss term is thus defined as:

$$\mathcal{L}_{(\text{Prop})} = \mathcal{L}_C(P_m^i, s_{(m)}^i, \{s_{(\neg m)}^i\}) + \mathcal{L}_C(P_m^i, s_{(m)}^i, \{s_{(m)}^{\neg i}\}) + \mathcal{L}_C(P_m^i, o_{pt(s_{(m)}^i)}, \{o_{\neg pt(s_{(m)}^i)}\}). \tag{13}$$

where $\mathcal{L}_C(p, \{pos\}, \{neg\})$ refers to the symmetric contrastive loss to force $p$ close to the positive samples $\{pos\}$ and far from the negative samples $\{neg\}$ (Radford et al., 2021). By optimizing Eq. (13), the representations of the visual proposals, sentences and prototypes on one hand will be aligned. Due to end-to-end learning, the abnormality localization is also learned in a weakly-supervised manner. Note that we drop the index $i$ in the following sections for the clarity.

**Loss for Self-critiquing:** To learn the self-critiquing mechanisms, we make use of triplet loss to push $s_{(m)}$ close to $P_m$ but apart from alternative proposals $C_m^{(\text{Abn})}$ and $C_m^{(\text{Img})}$ to improve retrieval reliability. The loss term is defined as:

$$\mathcal{L}_{(\text{Alt})} = \frac{1}{2\mathcal{M}} \sum_{m=1}^{\mathcal{M}} \big(\mathcal{L}_T(s_{(m)}, P_m, C_m^{(\text{Abn})}) + \mathcal{L}_T(s_{(m)}, P_m, C_m^{(\text{Img})})\big), \tag{14}$$

where $\mathcal{L}_T(a, pos, neg)$ is triplet loss with $a$ being the anchor, $pos$ the positive sample, and $neg$ the negative sample.

For false negative self-critiquing, we define an additional contrastive loss to guide $C^{(\text{Neg})}$ to be close to the sentences with abnormalities present but not hypothesized (positive mentions) $\{\hat{s}_{(j)}^+\}$, but far from those with abnormalities absent from being mentioned (negative mentions) $\{\hat{s}_{(j)}^-\}$:

$$\mathcal{L}_{(\text{Neg})} = \mathcal{L}_C(C^{(\text{Neg})}, \{\hat{s}_{(j)}^+\}, \{\hat{s}_{(j)}^-\}). \tag{15}$$

We train the proposed RadSCR model by optimizing: $\mathcal{L}_{(\text{Prop})} + \beta_1 \mathcal{L}_{(\text{Alt})} + \beta_2 \mathcal{L}_{(\text{Neg})}$ with the importance weights $\beta_1$ and $\beta_2$.

# 4 EXPERIMENT

**Data** We test the proposed RadSCR on three publicly available X-ray image-report datasets MIMIC CXR (Johnson et al., 2019a;b), ReXGradient (Zhang et al., 2025) and IU XRay (Demner-Fushman et al., 2016) for report generation, report retrieval and abnormality detection. We also use VinDR-CXR (Nguyen et al., 2022) to test the performance of abnormality localization on X-ray images.

**Baselines** For performance comparison, we evaluate a set of state-of-the-art approaches, including i) VLM-based approaches: `Transformer` (Vaswani et al., 2017), `R2Gen` (Chen et al., 2020), `R2GenCMN` Chen et al. (2021) `RGRG` (Tanida et al., 2023), ii) LLM-based: `Qwen3-VL (3B)` (Yang et al., 2025), `MedGamma (4B)` (Sellergren et al., 2025), `LLaVA-Med (7B)` (Li et al., 2023), `LLaVA-Rad (7B)` (Zambrano Chaves et al., 2025), `CoMT (7B)` (Jiang et al., 2025), and iii) retrieval-based approaches: `BiomedCLIP` (Zhang et al., 2023), `MedCLIP` Wang et al. (2022), `BioViL` (Boecking et al., 2022), `X-REM` (Jeong et al., 2023) and `CXR-RePaiR` (Endo et al., 2021). Related implementation details are reported in the Appendix A.1.

**Model Setting** We train RadSCR on MIMIC CXR (training set) and evaluate it by the test sets of MIMIC CXR, ReXGradient and IU XRay. We use the Swin Transformer (Liu et al., 2021) as visual extractor and Phi (4B) (Ren et al., 2025) as the LLM decoder with its parameters frozen. We consider $\mathcal{N} = 37$ abnormalities annotated by Chest ImaGenome Wu et al. (2021). The prototype number is set to $\mathcal{K} = 5$ for each abnormality, where sentences with positive mentions are clustered into 4 groups and those with negative mentions form the remaining one.

## 4.1 PERFORMANCE EVALUATION ON REPORT GENERATION

**Evaluation Metrics:** We evaluate the generated reports by i) `CheXbert` (Smit et al., 2020) of 14 observation accuracy, ii) Clinical Efficacy (CE) (Chen et al., 2020) extended to 37 abnormalities (`CE-Abn`) and the normality of 25 anatomical parts (`CE-Organ`), iii) `RadGraph-F1` (Jain et al., 2021) which also considers relationship correctness among observations, iv) `RadNLI` (Miura et al., 2021) which measures inference correctness of contradiction, entailment or neutral between generated reports and ground-truth, and v) `BLEU` (Papineni et al., 2002), `METEOR` (Banerjee & Lavie, 2005) and `ROUGE-L` (Lin, 2004) for measuring $n$-gram accuracy. The ground-truth annotations are used as targets in computing the metrics.

**Experimental Results and Discussion:** We conduct extensive experiments for performance evaluation based on MIMIC CXR, ReXGradient and IU Xray datasets. Table 1 shows the results on MIMIC CXR. Among baselines, LLaVA-Rad shows effective performance in diagnosing common chest abnormalities evaluated by `CheXbert`, and the region-based RGRG shows a high accuracy of reporting clinical observations relationship (occurrence and anatomical location) evaluated by `RadGraph-F1`. RadSCR outperforms other baselines in both clinical accuracy metrics `CheXbert` and `RadGraph-F1`. Meanwhile, RadSCR gives the best performance in the detection of anatomical abnormalities, as indicated by `CE-Abn` and `CE-Organ` scores, covering a wide range of chest abnormalities and their anatomical locations. In addition, RadSCR gives the highest `RadNLI` scores, indicating fewer diagnostic statements contradictory to the ground truth in the generated reports. These results suggest that RadSCR can effectively improve clinical accuracy in terms of both abnormality detection and diagnostic coherence to the ground-truth for report generation. Similar results are obtained for ReXGradient and IU Xray datasets (see Appendix A.1 for more details). We

| | Model | CheXbert | | CE | | RadGraph-F1 | | RadNLI | | |
|---|---|---|---|---|---|---|---|---|---|---|
| | | Acc. | F-1 | Abn. | Organ | P. | C. | Pr. | Re. | F-1 |
| VLM-based | Transformer | 0.201 | 0.304 | 0.208 | 0.269 | 0.191 | 0.130 | 0.161 | 0.217 | 0.135 |
| | R2Gen | 0.203 | 0.303 | 0.207 | 0.476 | 0.205 | 0.243 | 0.168 | 0.187 | 0.128 |
| | R2Gen-CMN | 0.157 | 0.402 | 0.258 | 0.416 | 0.201 | 0.137 | 0.144 | 0.199 | 0.109 |
| | RGRG | 0.383 | 0.489 | 0.251 | 0.669 | 0.321 | 0.248 | 0.379 | 0.326 | 0.317 |
| LMM-based | Qwen3-VL | 0.184 | 0.195 | 0.065 | 0.289 | 0.081 | 0.046 | 0.253 | 0.160 | 0.112 |
| | MedGamma | 0.419 | 0.413 | 0.219 | 0.407 | 0.141 | 0.086 | **0.469** | 0.143 | 0.150 |
| | LLaVA-Med | 0.397 | 0.135 | 0.041 | 0.555 | 0.202 | 0.139 | 0.332 | 0.335 | 0.312 |
| | LLaVA-Rad | 0.487 | 0.512 | 0.399 | 0.661 | 0.285 | 0.220 | 0.314 | 0.322 | 0.286 |
| | CoMT | 0.406 | 0.250 | 0.151 | 0.485 | 0.218 | 0.151 | 0.331 | 0.290 | 0.274 |
| Retrieval-based | BiomedCLIP | 0.309 | 0.221 | 0.184 | 0.675 | 0.235 | 0.175 | 0.335 | 0.314 | 0.305 |
| | BioViL | 0.403 | 0.367 | 0.325 | 0.595 | 0.232 | 0.173 | 0.300 | 0.302 | 0.274 |
| | MedCLIP | 0.032 | 0.297 | 0.106 | 0.153 | 0.112 | 0.071 | 0.175 | 0.25 | 0.161 |
| | CXR-RePaiR | 0.385 | 0.423 | 0.380 | 0.630 | 0.251 | 0.191 | 0.293 | 0.292 | 0.264 |
| | X-REM | 0.382 | 0.402 | 0.382 | 0.615 | 0.243 | 0.186 | 0.303 | 0.310 | 0.280 |
| | **RadSCR** | **0.574** | **0.610** | **0.572** | **0.744** | **0.422** | **0.367** | 0.440 | **0.433** | **0.408** |

Table 1: Performance comparison on report generation based on MIMIC CXR data. "P." and "C.": Partial and Complete correctness of observation relationship; "Pr." and "Re.": Precision and Recall.

| Model | CheXbert | | CE | | RadGraph-F1 | | RadNLI | | |
|---|---|---|---|---|---|---|---|---|---|
| | Acc. | F-1 | Abn. | Organ | P. | C. | Pr. | Re. | F-1 |
| RadFM | 0.566 | **0.635** | 0.545 | 0.652 | 0.399 | 0.367 | 0.432 | 0.401 | 0.395 |
| MAIRA-2 | **0.581** | 0.621 | 0.565 | 0.701 | **0.444** | **0.379** | **0.445** | 0.422 | **0.410** |
| **RadSCR** | 0.574 | 0.610 | **0.572** | **0.744** | 0.422 | 0.367 | 0.440 | **0.433** | 0.408 |

Table 2: Performance comparison on MIMIC CXR against some recent models with larger decoders.

further compared RadSCR with some more recent models with larger decoders, including MAIRA-2 (7B) (Bannur et al., 2024) and RadFM (Wu et al., 2025), as shown in Table 2. Although they have significantly larger model sizes, our proposed RasSCR still achieved comparable results.

**Ablation Study:** To better understand the impact of different components in RadSCR, an ablation study is conducted by removing (i) self-critiquing mechanisms in both training and testing, (ii) self-critiquing mechanisms in testing only, (iii) LLM decoder, and (iv) abnormality prototypes. The results are shown in Table 3. Referring to (i) and (ii), removing the self-critiquing mechanisms leads to obvious performance degradation. Meanwhile, the self-critiquing mechanism in inference can ensure generated reports of better quality (see (ii)). In addition, by eliminating $C^{(\text{Neg})}$, the scores of both `CheXbert(F-1)` and `CE-Abn` drop, indicating the lower accuracy of abnormality detection. It shows that the global features used by $C^{(\text{Neg})}$ could help RadSCR identify certain missed abnormalities. Furthermore, results in (iii) indicate drops in the `RadNLI` score when LLM is removed, indicating its role in ensuring the content coherence of the retrieved sentences and the generated report. Also, results in (iv) show that removing the prototypes leads to drops in `CE-Organ` and `RadGraph-F1` scores which also consider the accuracy of the associated anatomical parts. The use of prototypes allows sentences of the same abnormalities with context variations (e.g., regions of observation) to be better organized for more fine-grained retrieval.

**Effect of Sampling More Alternative Proposals:** We can extend the self-critiquing mechanisms by sampling more alternative proposals $C_m^{(\text{Abn})}$ and $C_m^{(\text{Img})}$. We first tested the number of alternatives randomly sampled $N_p$ from zero to three for both $C^{(\text{Abn})}$ and $C^{(\text{Img})}$, with the results shown in Fig. 2. $N_p = 0$ indicates that no alternative is considered. For cases with $N_p > 1$, the averaged effect of multiple sampled alternatives are computed for discounting the hypothesized proposal. By jointly increasing the number of $C^{(\text{Abn})}$ and $C^{(\text{Img})}$, the best performance is obtained at $N_p = 2$, where $2 \times \mathcal{M}$ alternative abnormalities and $2 \times \mathcal{M}$ alternative X-ray images are considered. While the result implies that considering multiple alternatives can improve the effectiveness of the self-critiquing mechanisms, the optimal ways of sampling the alternatives remain open. We tested also different schemes of controlling randomness based on the patients' metadata and achieved further performance improvement (see Appendix A.1 for more details).

| RadSCR w/o. | | CheXbert(F-1) | CE-Abn | CE-Organ | RadGraph(Complete) | RadNLI(F-1) |
|---|---|---|---|---|---|---|
| | - | 0.610 | 0.572 | 0.744 | 0.367 | 0.408 |
| (i) | $C^{(\mathrm{Img})}$ | 0.581 | 0.542 | 0.691 | 0.371 | 0.388 |
| | $C^{(\mathrm{Abn})}$ | 0.560 | 0.523 | 0.688 | 0.300 | 0.369 |
| | $C^{(\mathrm{Neg})}$ | 0.602 | 0.556 | 0.709 | 0.345 | 0.367 |
| | $C^{(*)}$ | 0.561 | 0.535 | 0.689 | 0.289 | 0.359 |
| (ii) | $C^{(\mathrm{Img})}$ | 0.605 | 0.450 | 0.669 | 0.311 | 0.325↓↓ |
| | $C^{(\mathrm{Abn})}$ | 0.577 | 0.465 | 0.648↓↓ | 0.253 | 0.335 |
| | $C^{(\mathrm{Neg})}$ | 0.491↓↓ | 0.545 | 0.668 | 0.354 | 0.398 |
| | $C^{(*)}$ | 0.545↓ | 0.379↓ | 0.653↓ | 0.231↓ | 0.343 |
| (iii) | LLM | 0.611 | 0.554 | 0.724 | 0.351 | 0.326↓ |
| (iv) | $\{O_k\}_{k=1}^{K}$ | 0.591 | 0.377↓↓ | 0.751 | 0.210↓↓ | 0.357 |

Table 3: Results of ablation study by MIMIC CXR. $C^{(*)}$ refers to $\{C^{(\mathrm{Img})}, C^{(\mathrm{Abn})}, C^{(\mathrm{Neg})}\}$. "↓↓" and "↓" indicate the scores with the largest and second largest drops, respectively.

In addition, other than random sampling, we further investigated the effectiveness of other strategic schemes of sampling alternatives, including sampling those most similar to the visual proposal (hard samples), sampling those least similar (easy samples), as well as sampling a mixture of them. As shown in Table 4, some strategic sampling schemes (e.g., Random+hard+easy) can further boost the performance for some metrics. In general, how to better sample the alternatives to achieve more effective self-critiquing is an interesting direction of extending the RadSCR's framework.

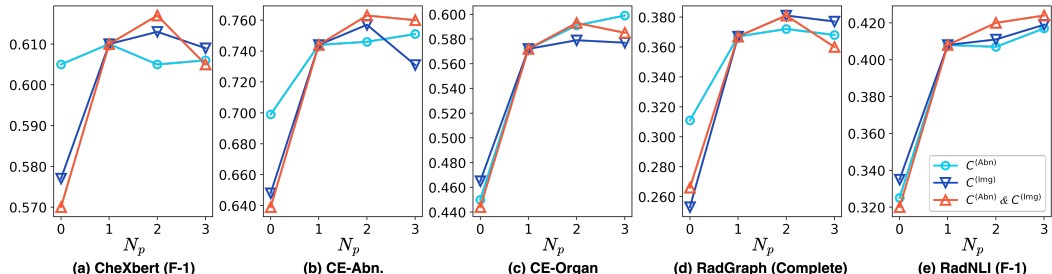

Figure 2: Performance of RadSCR with different numbers of alternative proposals sampled.

| Sampling | CheXbert | | CE | | RadGraph-F1 | | RadNLI | | |
|---|---|---|---|---|---|---|---|---|---|
| | Acc. | F-1 | Abn. | Organ | P. | C. | Pr. | Re. | F-1 |
| One alternative abnormality sampling | | | | | | | | | |
| Random | **0.574** | **0.610** | **0.572** | 0.744 | 0.422 | **0.367** | 0.440 | 0.433 | 0.408 |
| Only hard sample | 0.561 | 0.577 | 0.532 | 0.720 | **0.432** | 0.362 | **0.444** | 0.451 | **0.413** |
| Only easy sample | 0.568 | 0.593 | 0.566 | **0.751** | 0.426 | 0.361 | 0.432 | 0.412 | 0.399 |
| Three alternative abnormalities sampling | | | | | | | | | |
| Random | 0.570 | **0.609** | 0.577 | 0.731 | 0.441 | 0.377 | **0.448** | 0.455 | 0.419 |
| Random+hard+easy | **0.575** | 0.606 | **0.583** | 0.739 | **0.446** | **0.381** | 0.442 | **0.462** | **0.421** |

Table 4: Performance of different alternatives sampling schemes for critiquing on MIMIC CXR.

## 4.2 PERFORMANCE EVALUATION ON RETRIEVAL RESULTS

To further evaluate the ranking quality of the sentence retrieval results, we make use of *Accuracy-K* and some preference ordering metrics. The former gives the percentage of target sentences found in the top-*K* results. For the latter, we consider the top-50 results and measure the percentage of sentences with correct positive diagnoses ranking higher than the following three types of less preferred sentences: i) *Incomplete*: sentences with correct positive abnormalities but incomplete, ii) *Partially Correct*: sentences with correct and incorrect positive abnormalities, and iii) *Incorrect*: sentences without correct positive abnormalities. We denote the three preference ordering (PO)

metrics as PO-1, PO-2 and PO-3. As shown in Table 5, RadSCR performs consistently better than the baselines for highly-ranked sentences as indicated by `Acc@5` and `Acc@10`. It also achieves the best preference ordering scores, indicating its effectiveness in preserving overall sentence ranking.

| Model | IR Accuracy | | Preference Order | | |
|---|---|---|---|---|---|
| | Acc@5 | Acc@10 | PO-1 (Incomplete) | PO-2 (Partially Correct) | PO-3 (Incorrect) |
| CXR-RePaiR | 0.106 | 0.106 | 0.010 | 0.015 | 0.026 |
| BiomedCLIP | 0.266 | 0.288 | 0.478 | 0.514 | 0.467 |
| MedCLIP | 0.010 | 0.042 | 0.480 | 0.534 | 0.432 |
| BioViL | 0.171 | 0.171 | 0.523 | 0.528 | 0.523 |
| X-REM | 0.243 | 0.302 | 0.023 | 0.025 | 0.031 |
| **RadSCR** | **0.277** | **0.347** | **0.644** | **0.659** | **0.606** |

Table 5: Comparing the quality of sentence retrieval based on MIMIC CXR dataset.

### 4.3 PERFORMANCE EVALUATION ON ABNORMALITY PREDICTION AND LOCALIZATION

To illustrate the importance of introducing region-awareness for abnormality prediction, we create a baseline which uses again Swin Transformer (Swin) as visual extractor, followed by linear projection layers (`MLP`) instead of the fine-tuned RadSCR for abnormality prediction. As shown in Table 6, `Swin+RadSCR` outperforms `Swin+MLP` for all accuracy metrics, indicating that the proposed RadSCR with the region-awareness introduced can effectively improve the discriminative properties of the visual feature for abnormality prediction.

| Model | F-1↑ | FPR↓ | PR-AUC↑ | ROC-AUC↑ |
|---|---|---|---|---|
| Swin+MLP | 0.160 | 0.440 | 0.562 | 0.756 |
| **Swin+RadSCR** | **0.208** | **0.425** | **0.703** | **0.900** |

Table 6: Evaluation on abnormality prediction. **FPR** refers to false positive rate.

We also compare RadSCR's performance on abnormality localization with two existing weakly-supervised localization methods, including one based on the patch-based approach `TDIL` (Li et al., 2018) and another one based on the attention-based approach `HAM` (Ouyang et al., 2020). The evaluation is based on VinDR dataset, and the mean average precision score is adopted for the metrics. As shown in Table 7, with the threshold of the Intersection of Union (IoU) set as $0.1/0.3/0.5$, `Swin+RadSCR` can better localize the ground-truth abnormality regions by a large margin compared to the baselines (see Appendix for more details).

| Model | IoU(0.1) | IoU(0.3) | IoU(0.5) |
|---|---|---|---|
| TDIL | 0.125 | 0.095 | 0.077 |
| HAM | 0.134 | 0.102 | 0.081 |
| Swin+MLP | 0.210 | 0.054 | 0.012 |
| **Swin+RadSCR** | **0.308** | **0.199** | **0.101** |

Table 7: Evaluation of abnormality localization on VinDR dataset with annotations of abnormality regions. The available model weights of TDIL and HAAL used in this experiment are trained on ChestXray8 (Wang et al., 2017), and Swin+MLP/RadSCR are trained on MIMIC CXR.

## 5 CONCLUSION

We propose a novel Radiology Self-Critiquing Reporting model framework called RadSCR which learn multi-faceted mechanisms to self-reflect and verify the potential abnormality regions by constructing visual proposals of hypothesized abnormalities presented. The self-critiqued proposals are then integrated by a retrieval-based approach to generate reliable radiology reports, outperforming the SOTA report generation methods in terms of clinical accuracy and improved reliability of the located abnormality regions. *Limitation* The possibly false negative abnormalities are critiqued in the whole chest region on the X-ray images. Thus, the critique results on these critique results might not indicate any specific abnormality region to be localized.

## 6 ETHICS STATEMENT

The authors confirm that there are no i) human subjects or practices to data set releases, ii) potentially harmful insights, methodologies, or applications, iii) potential conflicts of interest or sponsorship, iv) discrimination/bias/fairness concerns, v) privacy or security issues, or vi) legal compliance. The research integrity issues (e.g., IRB, documentation, research ethics) are not applicable for this work.

## 7 REPRODUCIBILITY STATEMENT

To ensure the reproducibility of this work, the authors prepare the implementation details in the Appendix section, including i) *Data:* Three datatsets used in the experiments are publicly accessible where the download links are provided in Appendix A.1. The data pre-processing is referred to the baselines (Chen et al., 2020; 2021); ii) *Model Implementation:* The backbone modules of the proposed model are referred to the implementation provided by `huggingface.co` where the links of model structures and the pre-trained weights are provided in Appendix A.2; iii) *Baselines:* The implementation of the baselines are all referred to their official source codes and papers, where the links of pre-trained parameters of their model weights are provided in Appendix A.5. The results may have minor variations due to the different machines deployed; and iv) *Evaluation Metrics:* The implementation of the evaluation metrics are presented in Appendix A.4, where the evaluation details and the links of the open-source codes are provided.

## 8 ACKNOWLEDGMENT

This work is partially supported by the National Natural Science Foundation of China (62302413), the Health and Medical Research Fund (23220312), and the General Research Fund RGC/HKBU12202621 from the Research Grant Council.

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

## A APPENDIX

### A.1 EXPERIMENT RESULTS ON MORE DATASETS

We present the experimental results on MIMIC CXR dataset [1]. We also tested our proposed approaches with the baselines using the IU Xray [2] and ReXGradient [3] datasets (as shown in Table 8). As observed, our proposed RadSCR achieved SOTA performance in most clinical accuracy metrics in both datasets, demonstrating the effectiveness of using RadSCR in clinical application to generate accurate radiology reports.

We also evaluate the proposed approach and the baselines by natural language generation (NLG) metrics (`BLUE`, `METEOR` and `ROUGE`). The results are shown in Table 9. As observed, the proposed RadSCR achieves SOTA performances in MIMIC CXR and ReXGradient datasets. We noted that the `Transformer`, `R2Gen` and `R2Gen-CMN` obtain comparable performances in IU Xray data, which are trained by IU Xray data. Compared with the rest models, which are not fine-tuned in IU Xray, these three models could better learn the reporting styles maintained by the dataset itself, which results in better *n*-gram accuracy measured by NLG metrics.

**Effect of Controlling Randomness During Alternative Patient Sampling:** We provide additional evaluation results by using different selection strategies based on the available metadata in the testing stage. As shown in Table. 10, controlling the randomness during alternative patient sampling is beneficial. In future work, we will also address how to achieve controlled sampling for datasets without available metadata.

### A.2 MODEL IMPLEMENTATION

We use Swin Transformer (Base) (Liu et al., 2021)[4] as visual encoder and Clinical-BERT as language encoder (Yan & Pei, 2022)[5]. The input images are resized to $224 \times 224$ and split into $\mathcal{HW} = 49$ patches, while the dimension is set to $\mathcal{D} = 512$. The training epoch is set to $40$ with the learning rate set to $5e\text{-}5$ and the batch size set to $64$. The maximum length of a sentence is set to $60$ tokens. The important weights are setting for i) prototype similarity: $\alpha_1 = 0.4$, ii) alternative visual proposals $C^{(\text{Abn})}$ and $C^{(\text{Img})}$: $\alpha_2 = 0.5$, iii) self-critiquing loss $\mathcal{L}_{(\text{Alt})}$ of alternative proposals $\beta_1 = 0.4$, and iv) self-critiquing loss $\mathcal{L}_{(\text{Neg})}$ of complementary proposals $\beta_2 = 0.6$.

---

[1] `https://physionet.org/content/mimic-cxr-jpg/2.0.0`

[2] `https://www.kaggle.com/datasets/raddar/chest-xrays-indiana-university`

[3] `https://huggingface.co/datasets/rajpurkarlab/ReXGradient-160K`

[4] `https://huggingface.co/microsoft/swin-base-patch4-window7-224`

[5] `https://huggingface.co/emilyalsentzer/Bio_ClinicalBERT`

| | Model | CheXbert | | CE | | RadGraph | | RadNLI | | |
|---|---|---|---|---|---|---|---|---|---|---|
| | | Acc. | F1 | Abn. | Organ | P. | C. | Pr. | Re. | F1 |
| **IU Xray** | | | | | | | | | | |
| VLM-based | Transformer ♭ | 0.806 | 0.512 | 0.169 | 0.595 | 0.358 | 0.281 | 0.452 | 0.411 | 0.396 |
| | R2Gen ♭ | 0.788 | 0.418 | 0.171 | 0.600 | 0.306 | 0.236 | 0.453 | 0.408 | 0.396 |
| | R2Gen-CMN ♭ | **0.820** | 0.445 | 0.177 | 0.612 | 0.348 | 0.283 | **0.486** | 0.404 | **0.410** |
| | RGRG | 0.668 | 0.459 | 0.155 | 0.720 | 0.340 | 0.228 | 0.450 | 0.431 | 0.399 |
| LMM-based | Qwen3-VL | 0.394 | 0.099 | 0.032 | 0.354 | 0.129 | 0.073 | 0.355 | 0.220 | 0.215 |
| | MedGamma | 0.719 | 0.310 | 0.158 | 0.482 | 0.186 | 0.117 | 0.514 | 0.222 | 0.238 |
| | LLaVA-Med | 0.620 | 0.282 | 0.033 | 0.582 | 0.215 | 0.142 | 0.330 | 0.341 | 0.320 |
| | LLaVA-Rad | 0.801 | **0.518** | 0.324 | 0.717 | 0.290 | 0.204 | 0.326 | 0.318 | 0.309 |
| | CoMT | 0.395 | 0.135 | 0.005 | 0.701 | 0.284 | 0.190 | 0.484 | 0.337 | 0.365 |
| Retrieval-based | BiomedCLIP | 0.795 | 0.381 | 0.050 | 0.716 | 0.315 | 0.240 | 0.342 | 0.422 | 0.351 |
| | BioViL | 0.781 | 0.436 | 0.268 | 0.710 | 0.299 | 0.209 | 0.384 | 0.360 | 0.345 |
| | MedCLIP | 0.087 | 0.092 | 0.089 | 0.657 | 0.164 | 0.121 | 0.170 | 0.215 | 0.148 |
| | CXR-RePaiR | 0.741 | 0.385 | 0.274 | 0.672 | 0.258 | 0.188 | 0.344 | 0.339 | 0.321 |
| | X-REM ♭ | 0.778 | 0.464 | 0.255 | 0.732 | 0.309 | 0.233 | 0.357 | 0.397 | 0.350 |
| | **RadSCR** | 0.796 | 0.499 | **0.366** | **0.752** | **0.369** | **0.300** | 0.381 | **0.466** | 0.390 |
| **ReXGradient** | | | | | | | | | | |
| VLM-based | Transformer | 0.612 | 0.429 | 0.081 | 0.560 | 0.180 | 0.111 | 0.312 | 0.300 | 0.296 |
| | R2Gen | 0.601 | 0.455 | 0.090 | 0.567 | 0.178 | 0.117 | 0.322 | 0.319 | 0.310 |
| | R2Gen-CMN | 0.620 | 0.450 | 0.101 | 0.580 | 0.190 | 0.121 | 0.333 | 0.311 | 0.319 |
| | RGRG | 0.401 | 0.222 | 0.095 | 0.630 | 0.198 | 0.135 | 0.357 | 0.345 | 0.335 |
| LMM-based | Qwen3-VL | 0.368 | 0.170 | 0.060 | 0.298 | 0.080 | 0.047 | 0.271 | 0.197 | 0.163 |
| | MedGamma | 0.563 | 0.342 | 0.174 | 0.408 | 0.142 | 0.092 | **0.516** | 0.196 | 0.217 |
| | LLaVA-Med | 0.440 | 0.191 | 0.098 | 0.555 | 0.193 | 0.129 | 0.338 | 0.331 | 0.318 |
| | LLaVA-Rad | 0.626 | **0.478** | 0.326 | 0.680 | 0.228 | 0.167 | 0.340 | 0.313 | 0.309 |
| | CoMT | 0.653 | 0.143 | 0.665 | 0.070 | 0.235 | 0.653 | 0.494 | 0.315 | 0.349 |
| Retrieval-based | BiomedCLIP | 0.635 | 0.332 | 0.215 | 0.583 | 0.171 | 0.137 | 0.325 | 0.340 | 0.318 |
| | BioViL | 0.529 | 0.311 | 0.115 | 0.632 | 0.164 | 0.116 | 0.359 | **0.354** | 0.345 |
| | MedCLIP | **0.668** | 0.009 | 0.259 | 0.293 | 0.077 | 0.028 | 0.333 | 0.331 | 0.331 |
| | CXR-RePaiR | 0.517 | 0.357 | 0.156 | 0.585 | 0.192 | 0.143 | 0.336 | 0.333 | 0.323 |
| | X-REM | 0.547 | 0.398 | 0.157 | 0.598 | 0.213 | 0.167 | 0.346 | 0.344 | 0.332 |
| | **RadSCR** | 0.644 | 0.459 | **0.344** | **0.698** | **0.240** | **0.179** | 0.369 | 0.351 | **0.356** |

Table 8: Comparison of report generation by clinical accuracy metrics on IU Xray and ReXGradient data. Models with ♭ are tested using the official parameters pre-trained on the testing dataset.

| | Model | MIMIC CXR | | | IU XRay | | | ReXGradient | | |
|---|---|---|---|---|---|---|---|---|---|---|
| | | B. | M. | R. | B. | M. | R. | B. | M. | R. |
| VLM-based | Transformer | 0.115 | 0.160 | 0.287 | 0.231 | 0.360 | 0.402 | 0.099 | 0.121 | 0.192 |
| | R2Gen | 0.100 | 0.142 | 0.282 | 0.214 | 0.346 | 0.383 | 0.081 | 0.119 | 0.199 |
| | R2Gen-CMN | 0.132 | 0.210 | 0.302 | **0.244** | **0.398** | **0.414** | 0.085 | 0.212 | 0.233 |
| | RGRG | 0.154 | 0.328 | 0.365 | 0.128 | 0.333 | 0.380 | 0.091 | 0.255 | **0.264** |
| LMM-based | Qwen3-VL | 0.040 | 0.199 | 0.142 | 0.040 | 0.209 | 0.140 | 0.030 | 0.177 | 0.111 |
| | MedGamma | 0.037 | 0.206 | 0.144 | 0.036 | 0.209 | 0.140 | 0.028 | 0.185 | 0.118 |
| | LLaVA-Med | 0.111 | 0.231 | 0.243 | 0.090 | 0.213 | 0.224 | 0.079 | 0.220 | 0.201 |
| | LLaVA-Rad | 0.206 | 0.336 | 0.342 | 0.141 | 0.270 | 0.288 | 0.138 | 0.279 | 0.256 |
| | CoMT | 0.100 | 0.290 | 0.219 | 0.100 | 0.333 | 0.240 | 0.077 | 0.263 | 0.200 |
| Retrieval-based | BiomedCLIP | 0.152 | 0.266 | 0.269 | 0.129 | 0.258 | 0.257 | 0.094 | 0.197 | 0.186 |
| | BioViL | 0.168 | 0.300 | 0.289 | 0.188 | 0.333 | 0.317 | 0.144 | 0.263 | 0.241 |
| | MedCLIP | 0.090 | 0.237 | 0.168 | 0.120 | 0.266 | 0.233 | 0.075 | 0.229 | 0.163 |
| | CXR-RePaiR | 0.174 | 0.312 | 0.294 | 0.162 | 0.295 | 0.290 | 0.120 | 0.230 | 0.205 |
| | X-REM | 0.161 | 0.286 | 0.291 | 0.166 | 0.294 | 0.309 | 0.137 | 0.258 | 0.226 |
| | **RadSCR** | **0.344** | **0.460** | **0.452** | 0.176 | 0.311 | 0.312 | **0.157** | **0.290** | 0.259 |

Table 9: Comparison of report generation by NLG metrics on MIMIC CXR, IU Xray and ReXGradient data. "B.", "M." and "R." indicates BLEU, METEOR and ROUGE scores.

To obtain the saliency map of the abnormality classification, we use the XGrad-CAM (Fu et al., 2020) [6] to extract the class activation map from the visual encoder. The bounding box extraction is referred to the open-source code [7].

---

[6] https://github.com/jacobgil/pytorch-grad-cam

| | Sampling | CheXbert | | CE | | RadGraph-F1 | | RadNLI | |
|---|---|---|---|---|---|---|---|---|---|
| | Acc. | F-1 | Abn. | Organ | P. | C. | Pr. | Re. | F-1 |
| MIMIC CXR | | | | | | | | | |
| - | 0.574 | 0.610 | 0.572 | 0.744 | 0.422 | 0.367 | 0.440 | **0.433** | 0.408 |
| PA/AP | **0.588** | **0.615** | 0.566 | 0.759 | **0.429** | **0.370** | 0.435 | 0.421 | 0.399 |
| Posture | 0.570 | 0.613 | **0.579** | 0.762 | 0.427 | 0.369 | **0.451** | 0.429 | **0.415** |
| ReXGradient | | | | | | | | | |
| - | 0.644 | 0.459 | 0.344 | 0.698 | 0.240 | **0.179** | 0.369 | **0.351** | **0.356** |
| Sex | 0.639 | 0.460 | 0.347 | 0.685 | 0.237 | 0.171 | 0.365 | 0.348 | 0.352 |
| Age | **0.650** | **0.462** | **0.351** | 0.703 | **0.243** | 0.175 | **0.374** | 0.342 | 0.350 |

Table 10: Results of applying controlled sampling in alternative image critiquing.

We use Phi (4B) (Ren et al., 2025) as the LLM decoder [8]. The prompt used for LLM decoding is shown in Fig. 3.

The experiment is conducted with an Intel(R) Xeon Gold CPU (2.70GHz) and four sets of NVIDIA Tesla V100S GPU. The training / inference time is reported in Table. 11.

| Module | **Training** (s) | **Inference** (s) |
|---|---|---|
| Abnormality Region Localization | 0.311 | 0.193 |
| Self-Critiqued Sentence Retrieval | 0.463 | 0.300 |
| LLM-based Report Generation | - | 0.203 |

Table 11: The estimated time of training / inference per image.

$\mathcal{N} = 37$ abnormalities targeted by `RadSCR` are provided by Chest ImaGenome Wu et al. (2021) which are annotated on MIMIC CXR data: *Low lung volumes*, *Plreual effusion*, *Edema*, *Atelectasis*, *Opacity*, *Pneumonia*, *Calcification*, *Lung cancer*, *Lesion*, *Mass/nodule*, *Costophrenic angle blunting*, *Consolidation*, *Aspiration*, Hyperaeration, *Vascular redistribution*, *Emphysema*, *Interstitial lung disease*, *Scarring*, *Vascular congestion*, *Pneumothorax*, *Fluid overload/heart failure*, *Granuloma*, *Lobar/segmental collapse*, *Tube/line*, *Alveolar hemorrhage*, *Increased reticular markings/ild pattern*, *Infiltration*, *Enlargement*, *Medical device*, *Pericardial effusion*, *Mediastinal displacement*, *Mediastinal widening*, *Hernia*, *Tortuous aorta*, *Spinal degenerative changes*, *Bone deformity*, and *Bone fracture*.

## A.3 ABNORMALITY-AWARE RETRIEVAL REPOSITORY CONSTRUCTION

To retrieve the relevant report sentences for critiquing the visual proposal, we first construct $\mathcal{N} = 37$ sentence repositories of $\mathcal{N} = 37$ abnormalities. For MIMIC CXR data, we use annotations provided by the Chest ImaGenome Wu et al. (2021), where each sentence of the report in MIMIC CXR is annotated with the abnormalities and anatomical parts mentioned. For IU Xray and ReXGradient without annotations, we use a BERT-based text classifier to predict all the abnormalities mentioned and the positive abnormalities described in the report. This text classifier is trained by the Chest ImaGenome annotations on MIMIC CXR reports. We collect sentences of the same abnormality into the same sentence repository. Sentences with more than one abnormality mentioned can be collected in multiple repositories. Given that some reports might mention some but not all negative abnormalities, there could be no sentences for some abnormalities to be collected. In this case, we will supplement the sentences of non-mentioned abnormalities by some simple templates.

## A.4 EVALUATION METRIC

`CE-Abn` covers 37 abnormalities considered by RadSCR. We finetune a text classifier (SapBERT Liu et al. (2020)[9]) to predict whether these 37 abnormalities are mentioned as positive (ob-

---

[7]`https://github.com/batmanlab/AGXNet/blob/ee99ef199f1f96f7d0c35336935bd117664e733c/utils.py`
[8]`https://huggingface.co/microsoft/Phi-4-reasoning`
[9]`https://huggingface.co/cambridgeltl/SapBERT-from-PubMedBERT-fulltext`

```
{user}
## Instruction
You are a AI assistant specialised in radiology X-ray imaging topics. You are provided with two sets of the diagnostic
results and expected to summarized them and generate a comprehensive radiology report. Below are requirements for
the report generation

**REQUIREMENTS**
- There are two sentence sets, one is *Primary* set, another is *Secondary* set. All sentences in *Primary* set MUST BE
USED for summarizing, where the details are expected to be maintained. The repeated content can be omitted. The
conflicts across the sentences can be removed.
- The sentences in *Secondary* set are used for summarizing if they are not opposite the content of the sentences in
*Primary* set. To summarize the sentences, the details are expected to be maintained, the repeated content can be
omitted and the conflicts across the sentence can be removed.
- The sentence set are provided with the format of "{Abnormality}": "{Sentence}". {Abnormality} is the chest-related
abnormality in radiology domain, indicating the diagnostic target of the following sentence. {Sentence} is a sentence
extracted from the radiology report, describing the observation related to {Abnormality}.
- Just output the report directly. DO NOT add additional explanations or introduce in the answer unless you are asked to.

## Example
*Primary* set
{
        "{Example_Abnormality_{11}}": "{Example_Sentence_{11}}",
        "{Example_Abnormality_{12}}": "{Example_Sentence_{12}}",
        ...
}
*Secondary* set
{
        "{Example_Abnormality_{21}}": "{Example_Sentence_{21}}",
        "{Example_Abnormality_{22}}": "{Example_Sentence_{22}}",
        ...
}
Report to be generated:
{
        "Report": "{Example_Report}"
}

## Input
{
        "{Abnormality_{11}}": "{Sentence_{11}}",
        "{Abnormality_{12}}": "{Sentence_{12}}",
        ...
}
*Secondary* set
{
        "{Abnormality_{21}}": "{Sentence_{21}}",
        "{Abnormality_{22}}": "{Sentence_{22}}",
        ...
}

## Output
Report to be generated:

{assistant}
...
```

Figure 3: Prompting data example used for the LLM decoder of the proposed RadSCR.

served on X-ray image) in the given report. The labels are annotated Chest ImaGenome Wu et al. (2021).

`CE-Organ` covers 25 anatomical parts annotated Chest ImaGenome Wu et al. (2021), including: *Left lung*, *Right lung*, *Left upper lung zone*, *Right upper lung zone*, *Left mid lung zone*, *Right mid lung zone*, *Left lower lung zone*, *Right lower lung zone*, *Left hilar structures*, *Right hilar structures*, *Aortic arch*, *Cardiac silhouette*, *SVC*, *Cavoatrial junction*, *Upper mediastinum*, *Left costophrenic angle*, *Right costophrenic angle*, *Left clavicle*, *Right clavicle*, *Left apical zone*, *Right apical zone*, *Spine*, *Trachea*, *Left hemidiaphragm*, and *Right hemidiaphragm*. We finetune a text classifier (Sap-

BERT Liu et al. (2020)[10]) to predict whether there are any positive abnormalities associated with these 25 anatomical parts.

`CheXBert`[11] covers 14 common observations considered in Irvin et al. (2019): *Enlarged cardio-mediastinum*, *Cardiomegaly*, *Lesion*, *Lung opacity*, *Edema*, *Consolidation*, *Pneumonia*, *Atelectasis*, *Pneumothorax*, *Pleural effusion*, *Lung Other*, *Fracture*, *Support devices*, and *No Findings*.

`RadGraph`[12] covers 14,579 entities and 10,889 relations defined in the related work Jain et al. (2021).

`RadNLI`[13] covers the inference relationships of *Contradiction*, *Entailment* and *Neutral*.

Preference Order (`PO`) measures the correctness of 37 abnormalities considered in the RadSCR.

`BLUE`, `METEOE` and `ROUGE` are refer to a public-accessed implementation[14].

## A.5 BASELINE IMPLEMENTATION

`Transformer` composes an encoder of three-level transformer layers and a decoder of three-level transformer layers, as implemented by Chen et al. (2020)[15].

The remaining baselines are implemented according to their official codes and pre-trained weights, including R2Gen Chen et al. (2020): `https://github.com/cuhksz-nlp/R2Gen`; R2Gen-CMN Chen et al. (2021): `https://github.com/cuhksz-nlp/R2GenCMN`; RGRG Tanida et al. (2023): `https://github.com/ttanida/rgrg`; CXR-RePaiR Endo et al. (2021): `https://github.com/rajpurkarlab/CXR-RePaiR`, MdeCLIP Wang et al. (2022):`https://github.com/RyanWangZf/MedCLIP`, BiomedCLIP Zhang et al. (2023):`https://huggingface.co/microsoft/BiomedCLIP-PubMedBERT_256-vit_base_patch16_224`, BioViL Boecking et al. (2022):`https://github.com/martinzwm/biovil`, X-REM Jeong et al. (2023):`https://github.com/rajpurkarlab/X-REM`, TDIL Li et al. (2018) and HAM Ouyang et al. (2020):`https://github.com/oyxhust/HAM`.

## A.6 EVALUATION ON ABNORMALITY LOCALIZATION

VinDr-CXR Nguyen et al. (2022; 2020)[16] provides annotations of abnormality regions, including *Infiltration*, *Lung Opacity*, *Consolidation*, *Nodule/Mass*, *Pulmonary fibrosis*, *Pleural thickening*, *Aortic enlargement*, *Cardiomegaly*, *ILD*, *Other lesion*, *Pleural effusion*, *Calcification*, *Enlarged PA*, *Lung cavity*, *Atelectasis*, *Mediastinal shift*, *Lung cyst*, *Pneumothorax*, *Emphysema*, *Clavicle fracture*, *Rib fracture*, and *Edema*.

ChestXray8 Wang et al. (2017)[17] provides annotations of abnormality regions, including *Atelectasis*, *Cardiomegaly*, *Pleural effusion*, *Infiltrate*, *Mass*, *Nodule*, *Pneumonia*, and *Pneumothorax*.

We evaluated the localization results of the abnormalities shared across MIMIC CXR (Chest ImaGenome), VinDr-CXR and ChestXray8. We noted that the localization annotations of the same abnormality from different datasets could be variable to some extent, as the localization results of radiologists could be affected by population differences, local operating rules, or personal experiences. However, for common chest abnormalities, their relevant regions to be localized by different radiologists should be similar in most cases, as the related diagnostic consensus for these abnormalities has been studied for years.

---

[10]`https://huggingface.co/cambridgeltl/SapBERT-from-PubMedBERT-fulltext`
[11]`https://github.com/stanfordmlgroup/CheXbert`
[12]`https://github.com/Stanford-AIMI/radgraph`
[13]`https://github.com/Mayo-Clinic-RadOnc-Foundation-Models/Radiology-NLI`
[14]`https://github.com/salaniz/pycocoevalcap`
[15]`https://github.com/cuhksz-nlp/R2Gen`
[16]`https://vindr.ai/datasets/cxr`
[17]`https://nihcc.app.box.com/v/ChestXray-NIHCC`

## A.7    Visualization of Abnormality Localization

We present two cases of progressive change of saliency maps with abnormality region localized by RadSCR during the training process (as shown in Fig. 4 and Fig. 5). As observed, trained RadSCR could localize relevant regions of the abnormalities presented. Meanwhile, the localized regions also covers some irrelevant areas, which indicates that the weakly-supervised abnormality localization is still challenging.

## A.8    Latent Space Visualization

We present the visualization of concept embedding, prototype embedding, and sentence embedding of randomly sampled sentence sets learned with and without the proposed self-critiquing mechanism (as shown in Fig. 6 and Fig. 7). For each abnormality in every setting (with / without the proposed self-critiquing mechanism), we present the visualization of one concept embedding, $\mathcal{K} = 5$ prototype embeddings, and 100 randomly sampled sentence embeddings of which sentences are associated with each $k^{th}$ prototype (in total 500 sentence embeddings for each abnormality). We use the t-SNE algorithm to project the $\mathcal{D}$-dimensional embeddings into a 2-dimensional vector. In general speaking, the points of embeddings learned with the self-critiquing mechanism are gathered more than those without the self-critiquing mechanism in most plots. It might indicate that the critiqued embeddings could represent the related information of each abnormality with less irrelevant features. However, this kind of visualization is also affected by the dimension reduction algorithm we use, while it is not the only way to explain these visualization results. We also note that how to properly interpret the learned representation in latent space remains open.

## A.9    Trends of sentence similar scores of the visual proposal and the alternatives during training

The proposed RadSCR allows the visual proposal $P_m$ and the alternatives $(C_m^{(\text{Abn})}, C_m^{(\text{Img})})$ interact by contrasting their similarity scores with the sentence $s_{(m)}$ to compute a discounted similarity score $\tilde{\sigma}(P_m, s_{(m)})$ (according to Eq. 11). In addition, $C^{(\text{Neg})}$ further supplements $P_m$ to recover false negatives. As shown in Figure 8, during training, the similarity score between $P_m$ and $s_{(m)}$ increases while the scores between $(C_m^{(\text{Abn})}, C_m^{(\text{Img})})$ and $s_{(m)}$ decrease, as anticipated. In addition, the similarity score between false negatives $C^{(\text{Neg})}$ and the corresponding sentences $s_{(0)}$ increases, so that the learned model can detect missing abnormalities.

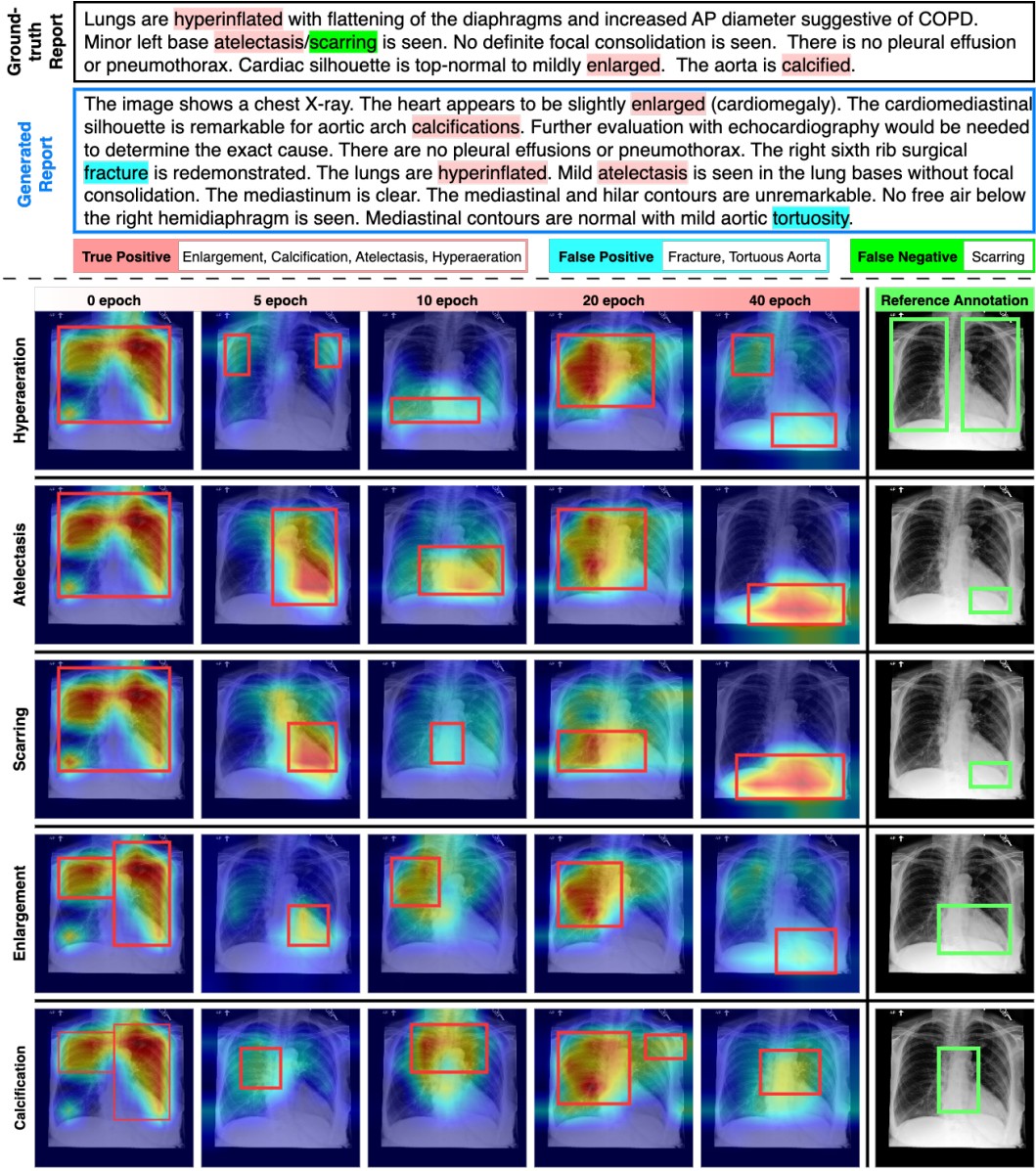

Figure 4: Illustration of progressive changing of saliency maps with abnormality localizations during the training process of RadSCR (Case I). The reference annotation of the abnormality regions (bounding boxes within green lines) are also provided which are inferred from the paired report.

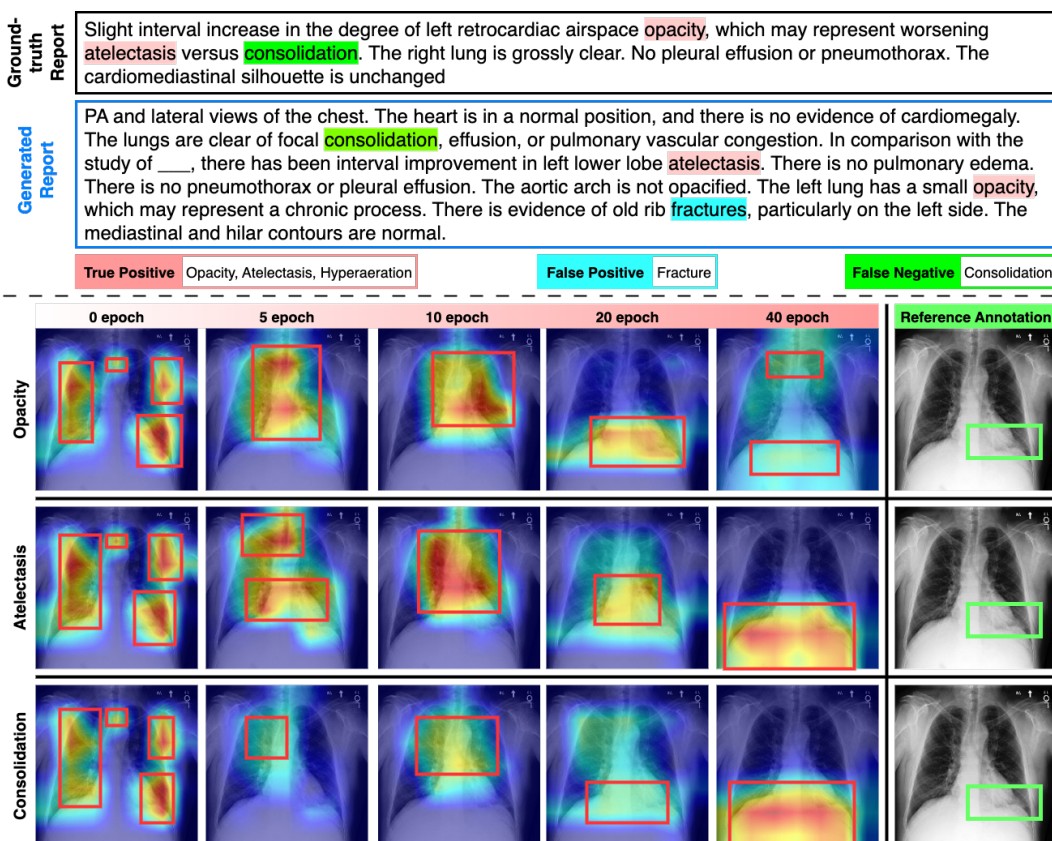

Figure 5: Illustration of progressive changing of saliency maps with abnormality localizations during the training progress of RadSCR (Case II).

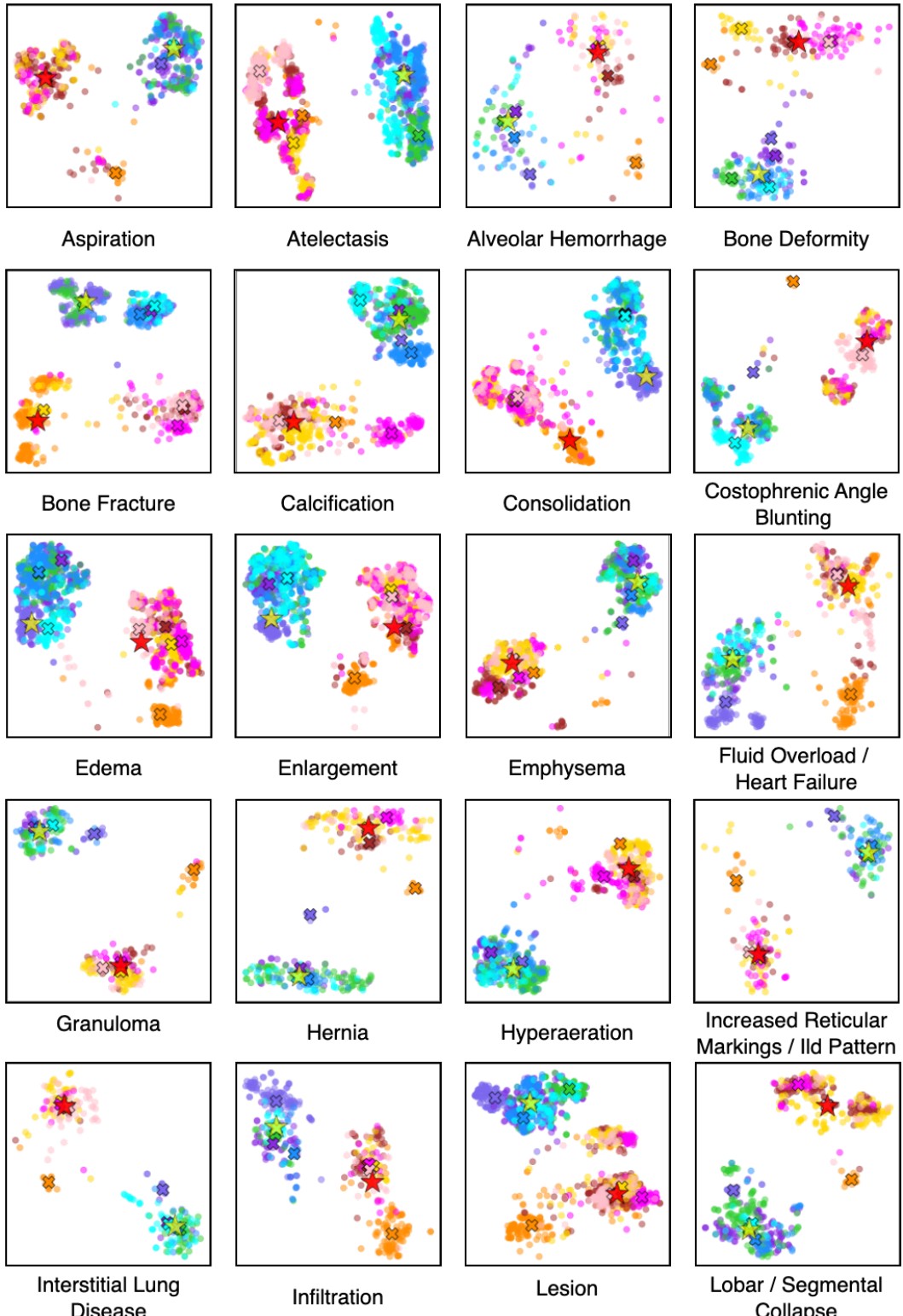

Figure 6: Visualization of concept embedding, prototype embedding and sentence embeddings in the latent space (Part I).

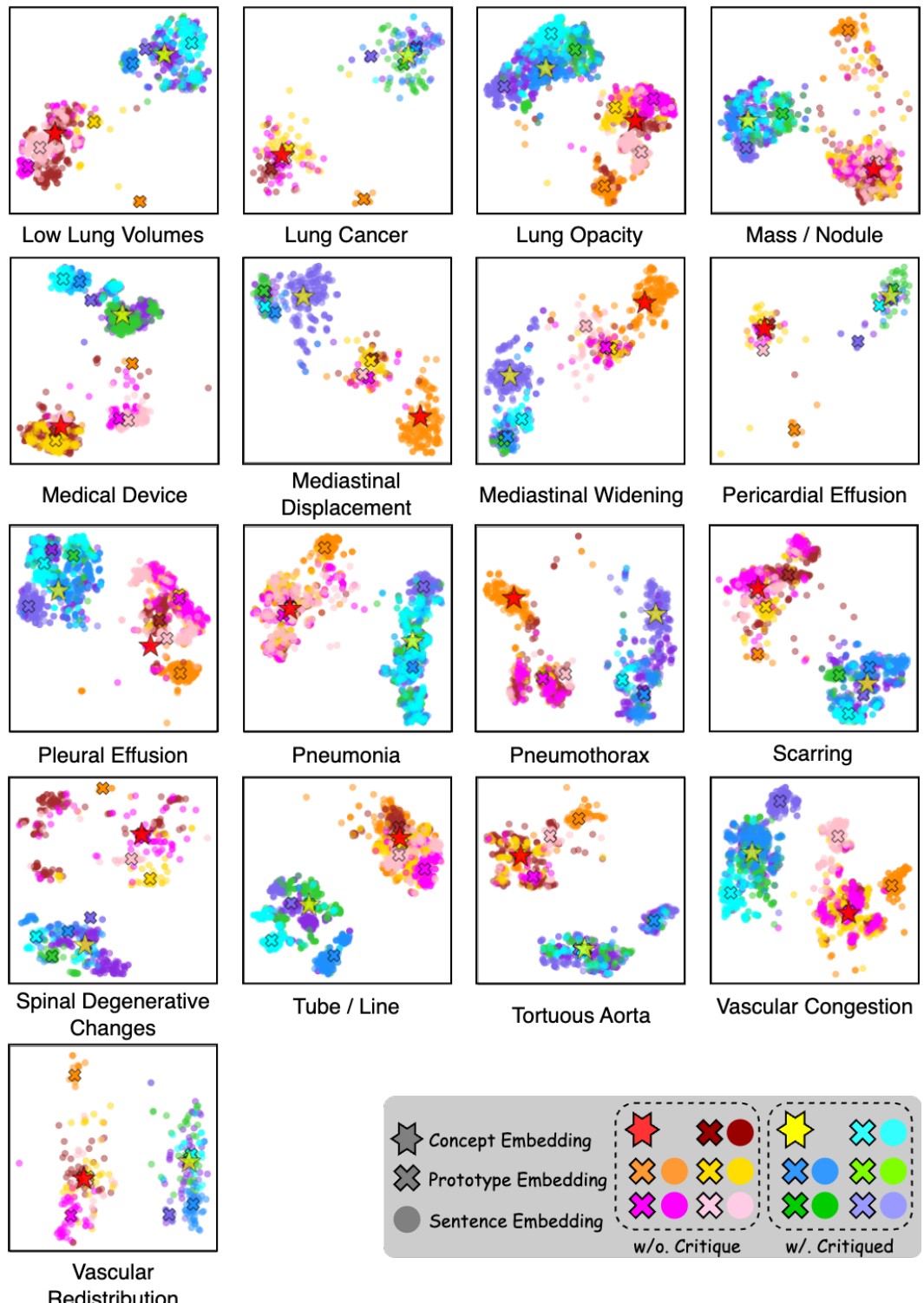

Figure 7: Visualization of concept embedding, prototype embedding and sentence embeddings in the latent space (Part II).

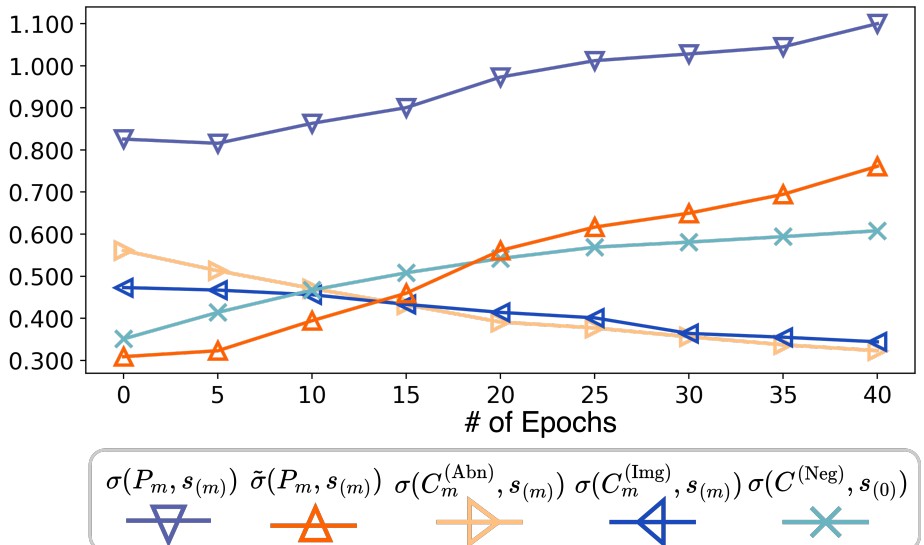

Figure 8: Trends of the sentence similar scores of the visual proposal and the alternatives during training. $\sigma(P_m, s_{(m)})$, $\sigma(C_m^{(Abn)}, s_{(m)})$, $\sigma(C_m^{(Img)}, s_{(m)})$, and $\sigma(C^{(Neg)}, s_{(0)})$ are the similarity scores defined in Section 3.3, and $\tilde{\sigma}(P_m, s_{(m)})$ is the overall discounted score.

