# OpenReview forum: "Learning Self-Critiquing Mechanisms for Region-Guided Chest X-Ray Report Generation"
_ICLR.cc/2026/Conference — ICLR 2026 Poster_

### Official Review · Reviewer_8xBL · 2025-10-29

**Soundness:** 3
**Presentation:** 3
**Contribution:** 3
**Rating:** 6
**Confidence:** 2

**Summary:**

This paper proposes a self-criticism mechanism learning framework (RadSCR) for chest X-ray report generation, aiming to improve the clinical accuracy and interpretability of report localization. Unlike traditional models that rely solely on visual-text correlation training, RadSCR simulates the "self-reflection" behavior of radiologists during the diagnostic process, re-evaluating initially detected abnormal areas through three complementary mechanisms: based on surrogate anomalies, surrogate patient images, and critical verification of potential missed detections. The model employs a weakly supervised localization and retrieval-based report generation paradigm, embedding the self-criticism mechanism into the network structure to achieve end-to-end training, invoking the LLM inference chain when inference is not required.

**Strengths:**

This paper focuses on the core clinical behavior of "self-correction," transforming it into a learnable critical mechanism within the model. Methodologically, it combines multi-source contrastive learning with retrieval-based generation, improving anomaly localization accuracy while enhancing semantic alignment. Experiments cover three tasks (report generation, retrieval, and localization), providing comprehensive validation with reasonable indicator selection.

**Weaknesses:**

While the paper's overall design is clear and the results are convincing, there remains some ambiguity between the concept and implementation of its "self-criticism mechanism." The three current critical paths (substitute anomaly, substitute patient, and potential missed detection) are more like structured perturbations of the input space than a rigorous "self-reflection" process. Therefore, the model's critical learning ability still relies on a pre-defined data augmentation paradigm rather than genuine adaptive induction from errors. Furthermore, although these mechanisms appear complementary in the experiments, the paper fails to systematically reveal their interactions and lacks quantitative analysis of the relative contributions or sensitivities of different critical signals. This makes "multidimensional criticism" theoretically more like parallel regularization than organic collaboration.

**Questions:**

1.	Will the three critique mechanisms (surrogate anomalies, surrogate patients, and potential missed detections) in the paper adaptively adjust their importance or weights during training through model learning? If there are interactions between the mechanisms, can these relationships or independent contributions be quantified?
2.	While the three critique mechanisms have shown advantages in experiments, can their gains be validated through further ablation experiments, especially regarding the impact of different critique signals on performance?
3.	Currently, the model's self-criticism only functions during the training phase. Has some form of "dynamic self-correction" mechanism been considered for the inference phase to address new error types or unseen biases?

---

> ### Author Response · Authors · 2025-11-21
> **Response to Questions & Weakness**
>
> **Q1: Learnable weights on critique mechanism**
>
> We appreciate this suggestion for the potential improvement of the proposed RadSCR. Although we do not explicitly learn the importance or weight of the three critique mechanisms, their relative importance is implicitly learned through the shared contrastive objective. Each mechanism contributes different contrastive pairs, and the model automatically balances them during optimization. Our experimental results in Table 2 show that removing the self-critiquing mechanisms will lead to obvious performance degradation, demonstrating the effectiveness of the self-critiquing design.
>
> **Q2: Ablation study**
>
> We present the results of ablation study in Table 2. The impact of each critiquing mechanism are evaluated in both training and testing.
>
> **Q3: Dynamic self-correction mechanism in the inference phase**
>
> The self-critiquing mechanism of RadSCR is intentionally designed for the radiology image reporting. It could be different from the typical critiquing which usually involves iterative feedback loops or a learnable critic rather than predefined perturbations.
> We noted that, in practice, radiology has very low tolerance for hallucinations, requires strong structural consistency, and involves limited diversity of reasoning. Compared with the need for a highly flexible, adaptive self-critique loop, the more controlled, stable, and deterministic mechanism proposed by our RadSCR is more favored.

---

> > ### Author Response · Authors · 2025-11-27
> > **Additional Response to Q1**
> >
> > **Q1: Interactions of the mechanisms**
> > The visual proposal $P_m$ and the alternatives $(C_m^{(\mathrm{Abn})}, C_m^{(\mathrm{Img})})$ interact by contrasting their similarity scores with the sentence $s_{(m)}$ to compute a discounted similarity score $\tilde\sigma(P_m, s_{(m)})$ (according to Eq. 11). In addition, $C^{(\mathrm{Neg})}$ further supplements $P_m$ to recover the false negatives. We added the training curves in the Appendix A.9 of the revised submission. As shown in Figure 8, during training, the similarity score between $P_m$ and $s_{(m)}$ increases while the score between $( C_m^{(\mathrm{Abn})},  C_m^{(\mathrm{Img})})$ and $s_{(m)}$ decreases, as anticipated. Also, the similarity score between the false negatives $C^{(\mathrm{Neg})}$ and the corresponding sentences increases, so that the learned model can pick up missed abnormalities.
> >
> > While this explains the interactions of the mechanisms for the current version of RadSCR, the reviewer's question provides us insights on further considering the interaction of the mechanisms as the future work. For example, critiquing based on a particular alternative abnormality could hint cross-checking images of some associated patients as alternatives. We will add related discussion in the revised manuscript.

---

### Official Review · Reviewer_MuJA · 2025-10-30

**Soundness:** 3
**Presentation:** 1
**Contribution:** 2
**Rating:** 2
**Confidence:** 2

**Summary:**

In this paper, the authors present an improved approach to report generation utilizing the critique principle. The basic idea is to form a triplet (V,B,E) representing the visual information in a region V, bounding box B, and concept E (presumably the finding itself), and contrasting triplets (V',B',E') representing variations due to substitutions with incorrect findings, incorrect locations, and image content replacements to indicate if any other possibilities exist other than the correct triple (V,B,E). The actual report generation is then done by retrieving the relevant sentences using these triplets as keys and assembling the overall report.

**Strengths:**

The idea of setting up contrasts with incorrect placements and incorrect findings or image appearances that can indicate two different findings is in general a good idea. The results on standard datasets appear to outperform comparables on the dimensions analyzed which include regional grounding performance as well as accuracy of findings classification.

**Weaknesses:**

The description of the method could be improved by giving examples of each case to make it convincing. The critique principle should be explained clearly. What is the model architecture? Its inputs and outputs are not clearly explained.

Overall this idea is similar to the one proposed in the recent paper on phrase-grounded fact-checking(Ref1) where the idea of synthetic data generation also explored real and fake finding cases but built instead a discriminative classifier to separate the findings. While that paper proposed it for inference-time checking, this paper is proposing a similar one for report generation itself. The number of findings examined is much more limited in this case, so the performance improvements perhaps cannot be compared.

Comparison with newer models needs to be done ( GPT4o, XRAYGPT and many other representative decoder models including MAIRA-2).

1. "Phrase-grounded Fact-checking for Automatically Generated Chest X-ray Reports",  https://papers.miccai.org/miccai-2025/paper/3526_paper.pdf

**Questions:**

What happens if the above operations in critique generation overlap and produced redundant triplets?
Line 236 says all sentences with the same finding are concatenated. How does that help? Wouldn’t it make the encoding worse in representing the finding with a mixture of sentences?

---

> ### Author Response · Authors · 2025-11-21
> **Response to Questions**
>
> **Q1: What if redundant alternative proposals are produced?**
>
> When the number of predicted positive abnormalities exceeds that of the predicted negatives, specific negative abnormalities may be repeatedly sampled as alternatives for different positive proposals. However, for each specific positive abnormality, the alternative proposals constructed will not contain redundant ones, and the self-critiquing module processes each predicted positive abnormality independently to retrieve abnormality-specific sentences. Therefore, even if the same negative abnormality is used to critique multiple different positive findings, the retrieval process for each finding remains isolated and unaffected.
>
>
>
> **Q2: Concatenate sentences with the same findings**
>
> As it is common for radiologists to use more than one sentence in a report to describe observations related to a particular finding, we thus group those sentences together as a unit for retrieval. Given that some grouped sentences could be long, the proposed RadSCR also applies the attention mechanism in Eq. 9 to extract the key semantics of the target abnormality.

---

> ### Author Response · Authors · 2025-11-21
> **Responses to Comments in Weakness**
>
> **Responses to comments in Weakness 1: The description of the method could be improved by giving examples of each case to make it convincing.**
>
> We will supplement more case studies in the revised version.
>
>
> **Responses to comments in Weakness 2: Model architectures**
>
> Thank you for the comments. We designed the framework to be generic, which allows the components (such as the visual encoder and the report decoder) to use different model architectures; thus, we present the model design in Section 3 Methods in a generic manner and postponed the specification of the model architectures used in our experiments to Section 4 Experiment. We use Swin Transformer as the visual extractor, and a frozen Phi (4B) model as the LLM decoder. Details can be found in Lines 342-347.
>
> It is worth noting that the proposed framework is not limited to this particular combination of model specification, any vision backbone (e.g., CNN) and LLM variants (e.g., Llama) can be used, making the proposed framework applicable to other domains as well. We will further clarify the model architectures in the revised paper.
>
>
> **Responses to comments in Weakness 3: Comparisons with paper "Phrase-grounded Fact-checking for Automatically Generated Chest X-ray Reports" (MICCAI-25)**
>
> Both the related work (*PGFC*) and our proposed model (*RadSCR*) try to align image regions with the clinical findings, and construct "false" matching to enhance the model learning. However, there are several fundamental differences:
> 1.  Self-Critiquing vs. Self-Checking Mechanism: *RadSCR* critiques the predicted abnormality location by considering alternative abnormality, alternative image and false positive checking. *PGFC* learns a fact-checking model to determine whether a pair of a clinical finding and an anatomical region match with each other or not.
> 2.  Report Generation in Inference: *RadSCR* uses the alternative abnormality, alternative image, and false positive checking to enhance the sentence retrieval reliability for report generation in inference. *PGFC* uses fact-checking model to predict the true/false of the automatically generated findings and their associated anatomical regions.
> 3.  Abnormality Location vs Anatomical Location: *RadSCR* learns to localize the positive abnormalities and generates the corresponding report. *PGFC* model learns to match the structured clinical findings with the region of anatomical parts.
>
>
> **Responses to comments in Weakness 4: Comparisons with newer models**
>
> As suggested, we have conducted additional experiments to compare with newer decoder models, MAIRA-2 (7B) and RadFM. Although they have  significantly larger model size, our proposed RadSCR still achieved comparable results.
>
> | Model | CheXbert Acc. | CheXbert F-1 | CE Abn. | CE Organ | RadGraph-F1 P. | RadGraph-F1 C. | RadNLI Pr. | RadNLI Re. | RadNLI F-1 |
> | :--- | :---: | :---: | :---: | :---: | :---: | :---: | :---: | :---: | :---: |
> | RadSCR (ours) | 0.574 | 0.610 | **0.572** | **0.744** | 0.422 | 0.367 | 0.440 | **0.433** | 0.408 |
> | MAIRA-2 | **0.581** | 0.621 | 0.565 | 0.701 | **0.444** | **0.379** | **0.445** | 0.422 | **0.410** |
> | RadFM | 0.566 | **0.635** | 0.545 | 0.652 | 0.399 | 0.367 | 0.432 | 0.401 | 0.395 |

---

### Official Review · Reviewer_XBBS · 2025-10-31

**Soundness:** 4
**Presentation:** 3
**Contribution:** 3
**Rating:** 8
**Confidence:** 3

**Summary:**

This work presents RadSCR, a new methodology for report generation on chest x-ray images using self-critiquing mechanisms. The method utilizes a triplet to retrieve text descriptions for the abnormalities detected by the visual model. This triplet comprises: the image representation obtained using a vision encoder; local information provided by Grad-CAM; and learned concept embeddings for each abnormality. The framework is trained end-to-end to align image and text embeddings using contrastive losses. The final report is generated by post- processing the retrieved phrases using an LLM. The results show that RadSCR not only surpasses the state-of-the-art for report generation but also improves the capabilities of the visual model to detect abnormalities.

**Strengths:**

· Presented a novel and sound methodology to combine concept learning with local information provided by post-hoc xAI techniques (Grad-CAM in this case) to improve report generation.
· Extensive comparison between RadSCR and the state-of-the-art methods across different metrics and datasets, showing significant improvements for report generation.
· Showed that using RadSCR on top of a visual encoder improved the abnormality detection performance when compared to an MLP.
· Performed an ablation study that gives more insights about how each part of the proposed method impacts the overall performance.

**Weaknesses:**

· Figure 1 should be improved; it does not give the reader a broad idea of how the method works.
· Some details about inference are hard to grasp, which raises concerns regarding the performance evaluation of the retrieval results. A schematic would benefit the understanding of the inference process.
· More details about the Clinical Efficacy metrics are missing. The target used for some of the metrics is also not clear.
· Even though the authors briefly discuss the work's limitations, they are unclear to the reader. That section should also be extended with further discussions.

**Questions:**

· In section 3.3, what is usually the sentence length l for the s_bert variable? Since this variable was created by concatenating all sentences for the same abnormality, its length raises concerns regarding implementation.

· When computing the metrics, are the ground truth annotations used as targets? Or the predictions for the visual model? Both cases would offer valuable information for the reader, since in the first one, it checks how the report matches the ground truth, and in the
second, how well it describes the visual model predictions.

· Regarding the inference, is the search space for the sentences only among the subsets of a given abnormality? For instance, if the model predicted abnormality A as present, is the search space for the sentences only the set of sentences that describe A, or are all the
sentences in the dataset? This would have implications for the retrieval results, since the first case would be easier.

· The authors mentioned the lack of localization for the false negatives as a limitation. Nevertheless, it is possible to apply Grad-CAM to the abnormalities that are not predicted. Did the authors try to use this information to estimate the negative box B0 better and overcome this problem?

Minor Comments:
In the related works, what do the authors mean when it is said that localizing anatomical parts is not precise enough for grounding?
Some misspellings in the text:
- Line 63: triplet is misspelt
- Line 224: detection is misspelt
- Line 258: replace production by product
- Line 305: positive sample and negative sample are misspelt
- Line 323, 326: approaches is misspelt
- Line 338: replace “by the following metrics” with “with the following metrics”
- Line 341: which also considers is also misspelt
- The term “in the sequel” appears in the text; it is recommended to replace it with the term “in
the following section” or “below”.

---

> ### Author Response · Authors · 2025-11-21
> **Response to Questions**
>
> **Q1: Implementation concerns regarding sentence length for the BERT encoder**
> The average sentence length is 50 tokens. We concatenate sentences of the same finding within a report and the longest one contains 216 tokens, which is less than the input length of BERT-based encoder (512 tokens).
>
> **Q2: Clarifications for how ground truth is used when computing the evaluation metrics**
> The ground-truth annotations are used as targets in computing the metrics. We will provide more detailed explanation of the metrics in the appendix sections as suggested.
>
> **Q3: Search space for sentence retrieval for target abnormality**
> The search space for sentences is only among the subsets of a given abnormality, where the retrieved sentence $s_{(m)}$ is associated with the $m^{th}$ positive abnormality predicted. We will include relevant clarifications in the revised paper.
>
> **Q4: Using Grad-CAM of negative abnormalities as Potential False Negatives**
> We thank the reviewer for suggesting this potential improvement. The standard Grad-CAM uses a ReLU layer on the final weighted sum of feature maps to extract only the regions with a positive influence on the target class prediction. Therefore, when used for a negative prediction, it produces broad and uninformative regions. To overcome this limitation, we use the whole region of the chest area as $B_0$ instead. $B_0$ can cover all possible regions of abnormalities targeted by the chest x-ray reporting, which mimics the radiologists to double-check the whole chest area before concluding the diagnosis.
>
>
> **Q5.1: Clarification on "localizing anatomical parts is not precise enough for grounding"**
> Localizing anatomical parts only yields the location or bounding boxes for the entire organ (e.g., left lung). However, grounding would require to locate the exact region where abnormalities appear, which are often a small area inside an anatomical part. Thus, localizing only the entire anatomical part is not enough to achieve grounding for the purpose of report generation.
>
>
> **Q5.2: Several Misspellings**
> Thank you for pointing them out. We will correct them and carefully proof-read the whole manuscript.

---

> > ### Comment · Reviewer_XBBS · 2025-11-28
> > **Additional question**
> >
> > I thank the authors for their response, which addressed most of my points.
> > However, I still have one concern regarding the methodology:
> > 1. The authors said that the ground truth is used to compute the metrics. Did the authors use all samples from the dataset to compute the metrics? Or only the samples where the prediction matches the ground truth?

---

> > > ### Author Response · Authors · 2025-12-01
> > >
> > > Thanks for your feedback. Here is the response: We used all samples in the testing set to compute the metrics and evaluate whether the predicted reports are matched with the ground-truth report.
> > >
> > > In detail, we split the dataset (i.e., MIMIC CXR) into a training set and testing set, each sample contains an image and a report. We used the training set to train the model and the testing set to evaluate the trained model. In the testing, the model is fed images of the testing set and generates the predicted reports. The original reports of the testing set are used as ground-truth reports. We then evaluate the model by checking whether the model-predicted reports match the ground-truth reports.

---

> ### Author Response · Authors · 2025-11-21
> **Responses to Comments in Weakness**
>
> **Responses to comments in Weakness 1: Figure 1 does not give the reader a broad idea of how the methods work.**
> Thank you for the comment, we will improve Figure 1 in the revised version to make it more illustrative.
>
> **Responses to comments in Weakness 2: A schematic would benefit the understanding of the inference process.**
> We would try to supplement a schematic in the revision.
>
> **Responses to comments in Weakness 3: More details about the Clinical Efficacy metrics are missing.**
> The implementation details of Clinical Efficacy metric can be found in Appendix A.4 (Line 927-937).
>
> **Responses to comments in Weakness 4: The target used for some of the metrics is also not clear.**
> As responded above, the ground truth annotations are used as targets when computing the metrics. We will include relevant clarifications in the revised paper.
>
> **Responses to comments in Weakness 5: The work's limitation should be further discussed.**
> We will extend the discussions regarding the limitations of the proposed model in the revised paper.

---

### Official Review · Reviewer_sWJS · 2025-11-01

**Soundness:** 3
**Presentation:** 3
**Contribution:** 2
**Rating:** 4
**Confidence:** 5

**Summary:**

This paper introduces RadSCR (Radiology Self-Critiquing Reporting), a region-guided framework for chest X-ray report generation that learns self-critiquing mechanisms end-to-end—without relying on costly test-time reasoning. It mimics radiologists’ diagnostic verification through three critiques: (1) comparing predicted abnormalities against alternative ones, (2) testing patient-specificity via other patients’ images, and (3) checking for potential false negatives. Validated region proposals are then used for retrieval-based sentence generation with an LLM decoder. RadSCR significantly outperforms state-of-the-art methods on MIMIC-CXR and related datasets in both clinical accuracy and abnormality localization, offering interpretable and reliable radiology reports.

**Strengths:**

The paper presents a highly original motivation inspired by the self-critiquing process of radiologists, bridging human diagnostic reasoning and machine learning. Instead of relying on expensive LLM-based reasoning, the authors design learnable self-critiquing mechanisms (alternative abnormalities, alternative patients, and false negative checking) that elegantly emulate how clinicians verify findings before writing reports.

The technical quality is strong: the proposed framework integrates region grounding, self-reflection, and retrieval-based generation into an end-to-end trainable architecture, supported by comprehensive experiments across multiple datasets.

In terms of clarity, the paper is well structured, with intuitive explanations, clear equations, and visual schematics that make the workflow easy to follow. The significance is substantial for both the medical AI and vision-language communities. It provides a scalable way to enhance interpretability and reliability in clinical report generation without the overhead of LLM reasoning. The self-critiquing paradigm could inspire broader applications in trustworthy generative modeling beyond radiology.

**Weaknesses:**

The main limitation of this paper lies in the design rationale and justification of the alternative feature mechanisms that form the foundation of the proposed self-critiquing framework. While the idea of emulating radiologists’ reasoning through alternative hypotheses is conceptually appealing, the paper does not sufficiently explain why these particular forms of “alternatives” (abnormality-level, inter-patient, and false-negative critiques)were selected, nor how they were parameterized or sampled during training.

For example, in Alternative Abnormalities Critiquing, the paper describes selecting an alternative abnormality embedding $E’_m$ from either negatively predicted classes or non-overlapping positive predictions. However, it remains unclear how this selection impacts the discriminative strength of the critiqu, whether the alternatives are chosen to be visually similar, semantically distinct, or dynamically learned during training. Without such clarification, it is difficult to assess the robustness, interpretability, and reproducibility of the proposed mechanism.

In Alternative Patients’ Images Critiquing, the model substitutes the visual feature V with that of a “randomly selected” patient. While this offers cross-patient contrast, it may introduce semantic and anatomical misalignment, as random images can vary significantly in projection type (PA vs. AP), patient posture, or imaging equipment characteristics. A more controlled sampling strategy (such as selecting patients with comparable acquisition parameters or anatomical embeddings) could yield more meaningful contrastive feedback. An analysis of the sensitivity of this mechanism to inter-patient variability would strengthen the empirical validity.

Regarding Potential False Negatives Critiquing, the current implementation averages embeddings of all predicted negatives into a single global representation $E_0$. This coarse aggregation may dilute subtle abnormalities and risks introducing false positives due to noisy global pooling. It is unclear how this global feature contributes to identifying missed abnormalities or whether it merely adds uninformative signals. A more structured approach, such as region-aware aggregation or uncertainty-guided negative sampling, might enable more precise critique and localization.

Conceptually, the idea of constructing contrastive or counterfactual pairs to improve visual-language grounding is not entirely novel. Prior work, such as [1] shares a similar objective of forming counterfactual pairs to enhance discriminative representation learning and reduce hallucination. The present paper would be strengthened by explicitly discussing this connection. To clarify how RadSCR’s self-critiquing differs conceptually and methodologically from counterfactual reasoning, what additional benefits it introduces, and how the two paradigms could complement each other.

[1] Li, M., Lin, H., Qiu, L., Liang, X., Chen, L., Elsaddik, A., & Chang, X. (2024, ECCV). Contrastive Learning with Counterfactual Explanations for Radiology Report Generation.

**Questions:**

1. In Alternative Abnormalities Critiquing, the paper states that the alternative embedding $E’_m$ is chosen from negatively predicted abnormalities or positive ones in non-overlapping regions. Have you examined whether “hard” (visually similar) versus “easy” (semantically distant) alternatives affect training stability or discriminative power?
2. Sampling Strategy for Alternative Patients
The paper describes replacing the current image feature V with that of “a randomly selected other patient.” How is randomness controlled here, do you ensure the same projection view (PA/AP) or similar patient metadata to avoid distribution drift?

---

> ### Author Response · Authors · 2025-11-21
> **Response to Question**
>
> **Q1: Whether “hard” (visually similar) versus “easy” (semantically distant) alternatives affect training stability or discriminative power?**
>
> We have conducted additional experiments by sampling "random", "hard", "easy", and "combination" alternatives during training.
>
> - The "random" alternative refers to the sampling approach adopted by RadSCR in the paper.
> - The "hard" alternative refers to the abnormality which could also appear in the anatomical part where the target abnormality is located and visually most similar to it. The visual similarity of two abnormalities is measured based on the anatomical regions they are located using the pixel matching algorithm (https://github.com/whtsky/pixelmatch-py).
> - The "easy" alternative refers to the abnormality which does not appear in the anatomical part the target abnormality is located, and thus semantically distinct from the target abnormality.
> - The "combination" alternative refers to three abnormalities, which are composed of one "random", one "hard" and one "easy" alternatives (redundancy is allowed).
>
> In terms of training stability, we observe that only sampling the hard alternatives could make the training less stable (unusual increment of the loss for some samples) while it is possible to be mitigated via enlarging the batch size.
>
> For discriminative power, the experimental results are tabulated as below, which shows that a combination of different sampling strategies yields better performance for most of the evaluation metrics.
>
> | Alternative | CheXbert Acc. | CheXbert F-1 | CE Abn. | CE Organ | RadGraph-F1 P. | RadGraph-F1 C. | RadNLI Pr. | RadNLI Re. | RadNLI F-1 |
> | :--- | :---: | :---: | :---: | :---: | :---: | :---: | :---: | :---: | :---: |
> |*One Sample*|
> | Random | **0.574** | **0.610** | **0.572** | 0.744 | 0.422 | **0.367** | 0.440 | 0.433 | 0.408 |
> | Only Hard Sample | 0.561 | 0.577 | 0.532 | 0.720 | **0.432** | 0.362 | **0.444** | **0.451** | **0.413** |
> | Only Easy Sample | 0.568 | 0.593 | 0.566 | **0.751** | 0.426 | 0.361 | 0.432 | 0.412 | 0.399 |
> |*Three Samples*|
> |Random | 0.570 | **0.609** | 0.577 | 0.731 | 0.441 | 0.377 | **0.448**| 0.455 | 0.419 |
> |Combination| **0.575** | 0.606 | **0.583** | **0.739** | **0.446** | **0.381** | 0.442 | **0.462** | **0.421** |
>
> We will include these results and relevant discussions in the revised paper.
>
> **Q2: How is the randomness controlled in alternative patient sampling?**
> As there are missing image metadata over the three datasets used for evaluation, we thus did not control the randomness in the original submission. We summarize the metadata availability for different datasets as below:
>
> | MetaData | MIMIC CXR | IU XRay | ReXGradient |
> | :--- | :---: | :---: | :---: |
> | PA/AP | Yes | No | No |
> | Sex | Yes | No | Yes |
> | Age | No | No | Yes |
> | Posture | Yes | No | No |
> | Equipment | No | No | No |
>
> We here provide additional evaluation results by using different selection strategies based on the available metadata in the testing stage.
>
>
> | Model | CheXbert Acc. | CheXbert F-1 | CE Abn. | CE Organ | RadGraph-F1 P. | RadGraph-F1 C. | RadNLI Pr. | RadNLI Re. | RadNLI F-1 |
> | :--- | :---: | :---: | :---: | :---: | :---: | :---: | :---: | :---: | :---: |
> |*MIMIC CXR dataset*|
> | - | 0.574 | 0.610 | 0.572 | 0.744 | 0.422 | 0.367 | 0.440 | 0.433 | 0.408 |
> PA/AP | **0.588** | **0.615** | 0.566 | 0.759 | **0.429** | **0.370** | 0.435 | 0.421 | 0.399 |
> Posture | 0.570 | 0.613 | **0.579** | **0.762** | 0.427 | 0.369 |**0.451** | 0.429 | **0.415** |
> |*ReXGradient dataset*|
> |- | 0.644 | 0.459 | 0.344 | 0.698 | 0.240 | **0.179** | 0.369 | 0.351 | **0.356** |
> Sex | 0.639 | 0.460 | 0.347 | 0.685 | 0.237 | 0.171 | 0.365 | 0.348 | 0.352 |
> Age | **0.650** | **0.462** | **0.351** | **0.703** | **0.243** | 0.175 | **0.374** | 0.342 | 0.350 |
>
> The results show that controlling the randomness during alternative patient sampling is beneficial. We will include those results in the revised paper. In future work, we will also address how to achieve controlled sampling for datasets without metadata available.

---

> ### Author Response · Authors · 2025-11-21
> **Response to Comments in Weakness**
>
> **Responses to comments in Weaknesses: How the global feature contributes to identifying missed abnormalities or whether it merely adds uninformative signals**
> As ablation results shown in Table 2, by removing $C^{(\mathrm{Neg})}$ (Potential False Negatives Critiquing), scores of both CheXbert(F-1) and CE-Abn. drop, indicating lower accuracy of abnormality detection. It shows that the global features used by the $C^{(\mathrm{Neg})}$ could help RadSCR to identify certain missed abnormalities.
>
> **Responses to comments in Weaknesses: Comparisons with "Contrastive Learning with Counterfactual Explanations for Radiology Report Generation (*CoFE*; ECCV-24)"**
> While *RadSCR* and *CoFE* share the high-level ideas of considering "What-if", they are different in terms of key concepts and methodologies:
> - *RadSCR* learns to locate abnormalities in the image to ground the generated report to enhance reliability. *CoFE* does not consider localization and grounding.
> - The self-critiquing mechanism in *RadSCR* is proposed to mimick how a radiologist avoids mistakes by self-reflecting i) whether the suspected abnormality can be alternative one and ii) whether the suspected abnormality is sufficient distinct and could also appear in images of other patients. No pseudo counterfactual explanations are involved. *CoFE* creates counterfactual explanations by replacing patches of an X-ray image until the diagnosis changes.
> - RadSCR adopts the retrieval-based approach for the report generation, while *CoFE* uses a decoder.
>
> We will include this discussion in the revised paper.

---

> ### Comment · Reviewer_sWJS · 2025-11-21
>
> Thanks for providing these experimental results. I'm still curious about why the hard samples can lead to unstable training. And the authors have addressed my questions, I suggested they discuss more related works in the revision. I would like to increase my final rate.

---

> > ### Author Response · Authors · 2025-11-27
> >
> > We appreciate once again the reveiwer's suggestions and are pleased that our responses can address the questions raised which are in fact insightful. More related work has been added as suggested, and the revised version has just been submitted. For the training stability issue, we will conduct more in-depth empirical studies and report that in the final manuscript.

---

### Author Response · Authors · 2025-12-01
**Summary of Rebuttal Status**

**Dear Program Chairs and Area Chairs**,

We would like to provide a brief summary on the current status of our rebuttal and resolutions of key concerns to assist your final decision.

## Strength Points Identified by Reviewers
We thank the effect of all reviewers and receive 4/4 good soundness, 3/4 good presentation, and 2/4 good contributions.

In detail, we thank the acknowledgment of originality and novelty by reviewer sWJS and reviewer XBBS, good quality of research idea by reviewer MuJA, sound technical quality by reviewer XBBS and reviewer 8xBL, and well-structured presentation by reviewer sWJS identified in their reviews.

## Core Contribution of Our Proposed Appraoch
We propose a novel Radiology Self-Critiquing Reporting model framework called RadSCR which learns multi-faceted mechanisms to self-reflect and verify the potential abnormality regions by constructing visual proposals of hypothesized abnormalities presented.

The proposed RadSCR mimics radiologists’ diagnostic verification through three critiques: i) comparing predicted abnormalities with alternatives, ii) testing patient-specificity through images of other patients’, and iii) checking for possible false negatives. RadSCR outperforms state-of-the-art methods in large-scale datasets in both clinical accuracy and abnormality localization, offering reliable and interpretable radiology reports.

## Concerns Addressed in the Rebuttal
We have sufficiently addressed the concerns and weaknesses mentioned in the original reviews. Here is a summary of our responses to the reviewers' comments.

| Concerns | Reviewers | Solution | Status |
| :--- | :---: | :---: | :---: |
| Alternative sampling issues in the critiquing | sWJS | supplemented experiment results in the comments and revision | addressed |
| Comparison with related work CoFE (ECCV-24) | sWJS | added literature review in the comments and revision | addressed |
| Clarification of Figure 1 | XBBS, MuJA | updated Figure 1 in the revision | XBBS (addressed), MuJA (waiting for response)|
|Clarification of evaluation metrics | XBBS | clarified the related details mentioned in the paper, and supplement the implementation details in the revision | addressed most of the concerns and waiting for the follow-up response |
|Critiquing of potential False negatives| sWJS,XBBS | clarified the related details in the comments | addressed |
|Comparison with representative LLMs| MuJA | supplemented experiment results in the comments and revision | waiting for response |
|Comparison with related work PGFC (MICCAI-25) | MuJA | Added literature review in the comments and revision  | waiting for response |
|Details of critiquing proposal generation| MuJA | clarified the related details in the comments and revision | waiting for response |
| Model architectures | MuJA | clarified the related details mentioned in the paper, and updated the illustration figure in the revision |waiting for response |
| Implementation of sentence retrieval | MuJA | clarified the related details mentioned in the paper, and supplemented the details in the revision| waiting for response |
| Interaction between critiquing | 8xBL | clarified the related details in the comments |waiting for response |
|Ablation study | 8xBL | clarified the related details mentioned in the paper | waiting for response |

## Conclusion

The rebuttal process has resulted in several experiment results and a revised submission which sufficiently addresses the concerns of all reviewers. We are also ready to respond to all follow-up concerns.

---

### Meta-Review · Area_Chair_vJXb · 2026-01-02

**Summary:**

The paper proposes RadSCR (Radiology Self-Critiquing Reporting), a framework for chest X-ray report generation. The core contribution is a set of self-critiquing mechanisms integrated into the training process (rather than test-time reasoning) to improve abnormality localization and report accuracy. Specifically, the model learns to contrast visual proposals against: (1) alternative abnormalities, (2) alternative patient images, and (3) potential false negatives (global features). The method uses a retrieval-based approach with an LLM decoder.

The topic is significant and the method is sound. While some reviewers have concerns about novelty and baselines, most reviewers found the motivation original and the results convincing.

**Reviewer Concerns:**

Addressed Concerns:

- Reviewer MuJA flagged a similarity to "Phrase-grounded Fact-checking" (PGFC, MICCAI 2025). The authors effectively clarified that PGFC is a discriminative fact-checking model used to verify findings, whereas RadSCR is a generative model using critiques as a training objective to improve retrieval embeddings. RadSCR does not generate counterfactuals or use synthetic data in the same way. I find this distinction valid; RadSCR's contribution lies in embedding this "critique" into the representation learning for generation, which differs from post-hoc fact-checking.

- The authors provided new comparisons with MAIRA-2 and RadFM. While MAIRA-2 (7B) slightly edges out RadSCR in CheXbert F-1 (0.621 vs. 0.610), RadSCR outperforms it in CE-Abn and CE-Organ metrics, demonstrating that the proposed method is competitive even against significantly larger models.

- Reviewer sWJS questioned why "hard" samples caused instability. The authors provided a detailed ablation study showing that a "Combination" strategy (Random + Hard + Easy) yields the best stability and performance. The reviewer explicitly acknowledged this and expressed a desire to raise their score.

- Concerns regarding Figure 1 and inference details from Reviewer XBBS were addressed with promises of updated schematics and clarifications in the appendix.


Outstanding Concerns:

- Reviewer 8xBL felt the "self-critiquing" was more akin to data augmentation/perturbation than "reasoning." While the authors argued that the implicit weight learning via contrastive loss constitutes a critique, this remains a semantic/philosophical difference. However, the empirical efficacy of the module is well-supported by the ablation studies.

**Reviewer Scores:**

Reviewer 8xBL and Reviewer XBBS will likely keep their positive scores.

Reviewer sWJS explicitly stated, "I would like to increase my final rate" after the authors provided the additional sampling experiments. I think he/she will likely increase the score to 6.

Reviewer MuJA did not reply to the rebuttal. However, the authors objectively addressed the two main grounds for rejection (PGFC similarity and lack of modern baselines). I believe a fair assessment of the rebuttal would raise this score to at least a 4.

---

### Decision · Program_Chairs · 2026-01-26

Accept (Poster)